# STT-LLM: Structural-Temporal Tokenization for Adapting LLMs to Longitudinal Profiles

## Abstract

Large Language Models have shown strong generalization across natural language tasks but remain underexplored for longitudinal biomedical profiles. In sports, biological profiles are analyzed for doping, with particular emphasis on two key challenges for longitudinal data: (i) sequence prediction for early detection of prohibited substance use, and (ii) anomaly detection for identifying doping-related deviations. We propose STT-LLM, a structural-temporal tokenization framework that adapts LLMs to longitudinal analysis without modifying the backbone architecture. STT-LLM constructs joint embeddings that capture both temporal dynamics and biological pathway-based interactions, which are then transformed into LLM-compatible tokens through the specialized structural and temporal tokenizers. We evaluate our approach on real-world longitudinal steroid datasets from athletes, where STT-LLM consistently outperforms LLM baselines. In addition, we present a case study where STT-LLM provides contextual reasoning that aligns more closely with expert assessments compared to baseline models. These results highlight the effectiveness of embedding-guided tokenization for adapting LLMs to understand longitudinal biological data.

## 1 Introduction

Large Language Models (LLMs) have shown generalization abilities across natural language and multimodal tasks, including reasoning and code generation (Chang et al., 2024; Matarazzo & Torlone, 2025). This led to growing interest in adapting LLMs to biological domains, particularly those with limited supervision and complex data structures. An example for analyzing longitudinal biological time-series is in the domain of sports doping control, where longitudinal profiles of athletes are analyzed to detect prohibited substance use. Longitudinal profiles are the instance (data representation) of a subject's trajectory over time, which are heterogeneous, irregularly sampled, and temporally dependent, often linked through metabolic pathways (Schüssler-Fiorenza Rose et al., 2019). Such structural and temporal dependencies are important for distinguishing natural physiological fluctuations from doping-induced abnormalities. However, most LLMs are pretrained on unstructured text corpora and rely on discrete token sequences, making them not well-suited for directly modeling numerical and time-dependent data (Raiaan et al., 2024; Naveed et al., 2023). Bridging this gap remains a non-trivial challenge, particularly in sports doping analytics. LLMs are promising: although their use in biological and clinical monitoring is still limited, recent studies show they can process clinical time-series data when augmented with specialized prompting or embedding methods (Chan et al., 2024; Xiao et al., 2025), suggesting that with appropriate adaptation, they can extend beyond language tasks to support longitudinal biomedical and sports monitoring applications.

*Challenge 1: Sequence prediction for early detection of prohibited substance use.* The longitudinal steroid profiles evolve along metabolic pathways, and deviations from expected trajectories may signal the administration of banned substances (Sottas et al., 2010). Anticipating such deviations requires sequence prediction that incorporates both temporal dynamics and pathway-level domain knowledge. However, standard LLMs operate on discrete text tokens (Naveed et al., 2023; Jia et al., 2025) and lack inductive biases to capture multivariate signals with irregular sampling. While prior works in graph learning (Ghanvatkar & Rajan, 2023; Luo et al., 2024) and time-series transformers (Tipirneni & Reddy, 2022; Xu et al., 2023) try to capture structural or temporal aspects, they rely on domain-specific architectures that cannot be directly integrated into general-purpose LLMs. This incompatibility limits their applicability in scenarios where flexible adaptation of pretrained LLMs is needed. Consequently,

without explicit structural-temporal modeling, LLMs struggle to align with the sequential dynamics required for early detection in doping monitoring.

*Challenge 2: Anomaly detection for doping identification.* Detecting doping abuse involves identifying subtle and rare deviations in metabolite concentrations that differ from natural physiological variation. These anomalies are often embedded in structural dependencies across metabolites, where perturbations in one metabolite cascade to others along metabolic pathways (Shukla & Marlin, 2018; Patharkar et al., 2024). Existing LLM approaches typically flatten profiles into text-like sequences or tabular inputs (Das et al., 2025), neglecting the relational structure necessary to capture biochemical constraints. While anomaly detection in time-series has been explored through specialized models (Lazaridou et al., 2021; Constantinou et al., 2023), these methods rely on domain-specific heuristics and cannot be directly integrated into generic LLMs. As a result, LLMs tend to default to trivial "normal" predictions, missing rare but important doping cases.

*Challenge 3: Limited and irregular (heterogeneity) athlete profiles.* Anti-doping laboratories have access to only one or two samples from most of the athletes, collected at irregular intervals due to testing constraints (Lauritzen & Solheim, 2024). This data scarcity makes it challenging to train robust supervised models, particularly given the high inter-individual variability in baseline steroid levels. While parameter-efficient fine-tuning methods (Han et al., 2024; Zhang et al., 2025) mitigate computational costs, they do not address the token mismatch between LLM inputs and structured biomedical signals. These limitations highlight the need for methods that enable LLMs to generalize under extreme data scarcity while preserving structural priors.

To address these challenges, we introduce a structural-temporal tokenization framework that adapts LLMs to analyze longitudinal biomedical profiles in sports doping. STT-LLM first constructs joint embeddings that capture both pathway-level structural dependencies among different metabolites and their irregular temporal dynamics, and then uses specialized structural and temporal tokenizers to transform these embeddings into LLM-compatible tokens. This design enables pre-trained LLMs with low-rank adaptation to perform domain-specific tasks such as sequence prediction for early detection of prohibited substance use and anomaly detection for doping identification.

**Key Contributions:**

- We propose STT-LLM, a structural-temporal tokenization framework that incorporates pathway structure and temporal dynamics of longitudinal profiles into specialized embeddings.

- We design tokenizers that convert these structural-temporal embeddings into LLM-compatible tokens, enabling efficient adaptation of pre-trained LLMs.

- We apply our approach to doping analysis, where STT-LLM outperforms LLM baselines in both sequence prediction and anomaly detection tasks under real-world conditions.

## 2  RELATED WORKS

**Tokenization and Embedding for Domain Adaptation**  Tokenization plays a foundational role in aligning raw inputs with the internal representations of LLMs, yet it remains a relatively underexplored area in domain adaptation compared to pretraining and fine-tuning strategies. Classical methods such as byte-pair encoding (Sennrich et al., 2015) and WordPiece (Wu et al., 2016) are effective for natural language but poorly suited for longitudinal biomedical data, where tokens should capture both structural dependencies (e.g., metabolic pathways) and irregular temporal dynamics. Recent efforts have explored task-aware tokenization for domain generalization and efficiency (Huang et al., 2025; Liu et al., 2024a). TAPEX (Liu et al., 2021a) and TabLLM (Hegselmann et al., 2023) adapt LLMs to tabular inputs through specialized token formats and training objectives, but these approaches fail to capture dynamic dependencies across time. In graph-based domains, GraphPrompt (Sun et al., 2022) and Graph-of-Thought (Besta et al., 2024) integrate relational structure via prompts or fusion modules, enabling zero-shot reasoning over interconnected data. Embedding strategies range from mean-pooling and distance-based transfer (Liu et al., 2021b) to hypernetwork-based token generation (Feher et al., 2024), but typically rely on auxiliary models or task-specific heuristics. While these methods highlight the importance of token design, they are not directly applicable to doping monitoring, where representations should jointly encode pathway-level structure and temporal progression.

**LLMs for Longitudinal Modeling** Recent work has explored repurposing LLMs for general time-series tasks through prompt augmentation and embedding reprogramming strategies (Rahman et al., 2024). For example, models such as Time-LLM (Jin et al., 2023) and UniTime (Liu et al., 2024b) demonstrate that pretrained LLMs can be reprogrammed to model time-indexed data by projecting temporal patches into token sequences. Despite these advances, most frameworks treat longitudinal signals as flat or fully textified inputs, neglecting the temporal granularity and variable semantics required in domains like anti-doping, where small deviations in steroid levels can have significant interpretive consequences. In related biomedical contexts, forecasting has been approached through timeline extraction (Frattallone-Llado et al., 2024) and event ordering (Leeuwenberg & Moens, 2020), but these methods often rely on fixed annotation spans and lack fine-grained temporal resolution. Traditional structured modeling in longitudinal data has relied on hand-crafted features or physiological scores (e.g., SOFA, SAPS) (Hou et al., 2020; Noroozizadeh et al., 2023), while more recent models integrate narrative texts and structured lab measurements (Jeong et al., 2024; Belyaeva et al., 2023). However, the gap between general-purpose LLMs and domain-specific data distributions remains a central challenge, particularly in doping monitoring, where datasets are highly individual-specific and require sensitivity to rare anomalous events under zero- or few-shot settings.

## 3 METHODOLOGY

### 3.1 PROBLEM FORMULATION

Let us consider a longitudinal profile consisting of repeated measurements of different clinical parameters across time. Formally, the longitudinal profile for a given athlete can be represented as $\mathbf{X}_i = [\mathbf{x}_{ij}] \in \mathbb{R}^{p \times n_i}$, where $p$ is the number of parameters, $n_i$ is the number of samples in profile $\mathbf{X}_i$, and $\mathbf{x}_{ijk}$ denotes the parameter $k$ of the $j^{th}$ sample. The longitudinal profile includes structural information encoded as a feature interaction graph $A \in \mathbb{R}^{p \times p}$, where $A_{k,l}$ represents the relationships between parameter $k$ and $l$. In this work, we address the following two main tasks:

**Sequence Prediction** Given the longitudinal profile up to time $t$, denoted as $\mathbf{X}_{i,1:t} = [\mathbf{x}_{ij}]_{j=1,\dots,t}$, the future samples are aimed to be predicted for $t+1$ as $\hat{\mathbf{x}}_{i,t+1} = f_\theta(\mathbf{X}_{i,1:t}, A)$, where $f_\theta$ is a predictive function parameterized by $\theta$. The function $f_\theta$ models both temporal dependencies across time and structural dependencies among parameters.

**Anomaly Detection** Irregular patterns in the longitudinal profile can be identified at two levels: i) *Local anomaly detection* to identify anomalous samples within an individual profile, meaning that one or more samples $\mathbf{x}_{ij}$ may show abnormal behavior relative to the athlete's own trajectory. Let us consider for each sample, a local anomaly score is computed $s_{ij}^{\text{local}} = g_\phi^{\text{local}}(\mathbf{x}_{ij}, \hat{\mathbf{x}}_{ij})$, where $\mathbf{x}_{ij} = [\mathbf{x}_{ij,1}, \dots, \mathbf{x}_{ij,p}]$, $\hat{\mathbf{x}}_{ij} = [\hat{\mathbf{x}}_{ij,1}, \dots, \hat{\mathbf{x}}_{ij,p}]$, and $g_\phi^{\text{local}}$ is a scoring function parameterized by $\phi$. One or more samples can be flagged as locally anomalous if their scores exceed a predefined threshold $\mathcal{A}_{\text{local}} = \{j \mid s_{ij}^{\text{local}} > \epsilon_{\text{local}}\}$. ii) *Global anomaly detection* to determine whether the entire longitudinal profile of an athlete is anomalous. Specifically, a profile is considered globally anomalous if any sample (preferably the recent sample collected for doping testing) is identified as anomalous. The global anomaly score is defined as $s_i^{\text{global}} = s_{i,n_i}^{\text{local}}$, where $n_i$ denotes the last sample index in the profile. A profile is classified as globally anomalous if $s_i^{\text{global}} > \epsilon_{\text{global}}$.

### 3.2 STT-LLM: STRUCTURAL-TEMPORAL TOKENIZATION FOR LARGE LANGUAGE MODELS

We propose STT-LLM (Fig. 1), which integrates joint structural-temporal embeddings, structural and temporal tokenizers to effectively capture and represent the intricate structural and temporal relationships inherent in longitudinal clinical profiles.

#### 3.2.1 INPUT PROMPT

The input prompt $\mathbf{I}$ consists of two components: the task $\mathbf{P}$, which is a textual description providing instructions, and the longitudinal profile $\mathbf{X}_i$. The task prompt $\mathbf{P}$ is processed using a pre-trained language tokenizer to produce token embeddings $Z_{\text{Pre}}$, while $\mathbf{X}_i$ is fed into the proposed tokenization

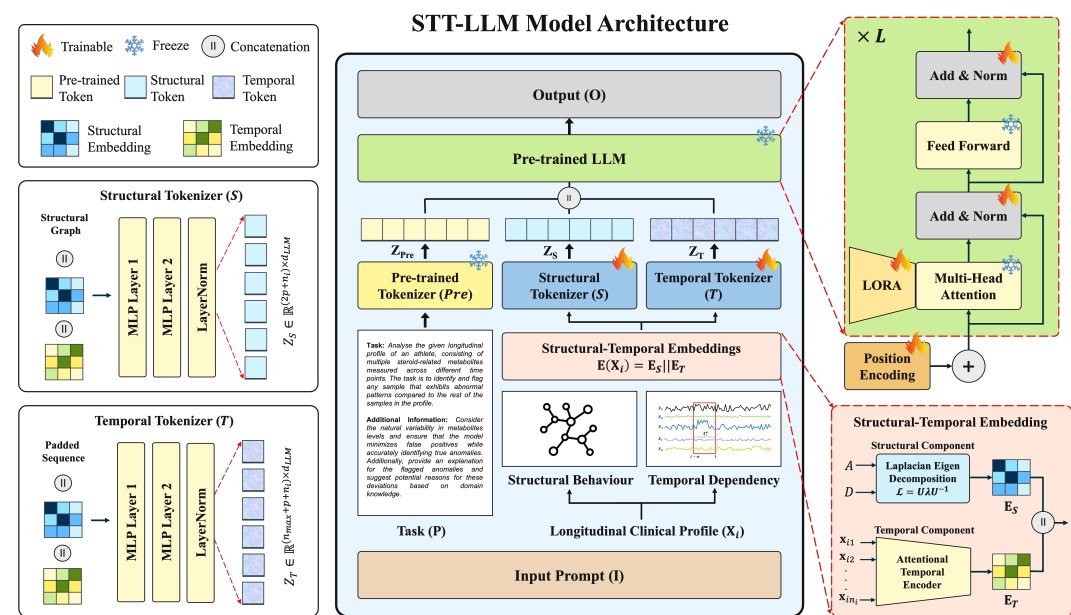

Figure 1: Proposed model architecture of STT-LLM for analyzing longitudinal clinical profile.

framework that integrates structural and temporal dependencies. This dual processing strategy enables the model to align semantic task instructions with rich domain-specific data representations.

### 3.2.2 STRUCTURAL-TEMPORAL EMBEDDINGS

**Structural Component** Given an adjacency matrix $A$ and a degree matrix $D$ of feature interaction graph, the normalized graph Laplacian $\mathcal{L} = I - D^{-\frac{1}{2}} A D^{-\frac{1}{2}}$ ($I$: identity matrix, $D$: node degrees) (Kipf & Welling, 2017). This normalized Laplacian $\mathcal{L}$ encodes important structural properties such as connectivity and community structure. The eigen-decomposition is calculated as $\mathcal{L} = U \lambda U^{-1}$ ($U$: eigenvectors, $\lambda$: eigenvalues). To obtain the structural embedding, the eigenvectors are projected through a learnable transformation: $\mathbf{E}_S = W_{\mathbf{E}_S} U + b_{\mathbf{E}_S}$ ($W_{\mathbf{E}_S}, b_{\mathbf{E}_S}$: trainable parameters).

**Temporal Component** The temporal behavior in the longitudinal profile is modeled using an attention mechanism as $\text{Attention}(Q, K, V) = \text{softmax}\left(\frac{QK^T}{\sqrt{d_k}}\right) V$, where $Q$, $K$, $V$ are linear projections (Vaswani et al., 2017). To incorporate temporal order, positional encodings are added, defined as $PE_{(pos, 2i)} = \sin\left(\frac{pos}{10000^{2i/d}}\right)$ and $PE_{(pos, 2i+1)} = \cos\left(\frac{pos}{10000^{2i/d}}\right)$ ($pos$: position, $i$: dimension index). These encodings allow the model to distinguish between positions in the input sequence. The attention output $\mathbf{A}_T$, is passed through a feed-forward network to produce $Z_{ST} = \text{ReLU}(\mathbf{A}_T W_{\mathbf{E}_{T_1}} + b_{\mathbf{E}_{T_1}}) W_{\mathbf{E}_{T_2}} + b_{\mathbf{E}_{T_2}}$ and layer normalization is applied to produce $\mathbf{E}_T = \text{LayerNorm}(Z_{ST})$. This architecture stabilizes training and facilitates gradient flow. The resulting temporal embeddings $\mathbf{E}_T$ capture dynamic patterns and dependencies important for modeling longitudinal profiles. Finally, the structural and temporal embeddings are concatenated to form the unified structural-temporal embedding $\mathbf{E}(\mathbf{X}_i) = \mathbf{E}_S \,||\, \mathbf{E}_T \in \mathbb{R}^{(p+n_i) \times p}$. This joint embedding ensures comprehensive integration of structural and temporal information, preparing the longitudinal clinical data for tokenization.

### 3.2.3 TOKENIZATION

**Structural Tokenizer** ($S$) The framework processes the structural aspects of the structural-temporal embeddings by effectively encoding a feature interaction graph constructed from domain knowledge in a longitudinal profile. The input structural representation $A$ of longitudinal profile is combined with the learned structural-temporal embedding $\mathbf{E}(\mathbf{X}_i)$, yielding the concatenated input:

$$\mathbf{X}_S = A || \mathbf{E}(\mathbf{X}_i), \quad \mathbf{X}_S \in \mathbb{R}^{(2p+n_i) \times p} \tag{1}$$

The concatenated input $\mathbf{X}_S$ is then processed through a multi-layer perceptron (MLP) with two layers. The first layer applies a ReLU nonlinearity $H_S = \text{ReLU}(\mathbf{X}_S W_{S_1} + b_{S_1})$, $H_S \in \mathbb{R}^{(2p+n_i) \times d_{\text{hidden}}}$, followed by a linear transformation $Z_S^{MLP} = H_S W_{S_2} + b_{S_2}$, $Z_S^{MLP} \in \mathbb{R}^{(2p+n_i) \times d_{LLM}}$, where $W_{S_1} \in \mathbb{R}^{p \times d_{\text{hidden}}}$, $W_{S_2} \in \mathbb{R}^{d_{\text{hidden}} \times d_{LLM}}$ ($b_{S_1}$, $b_{S_2}$: trainable parameters). To ensure stable training and consistent scaling of the token embeddings, layer normalization is applied $Z_S = \text{LayerNorm}(Z_S^{MLP})$, $Z_S \in \mathbb{R}^{(2p+n_i) \times d_{LLM}}$, where $d_{LLM}$ is the target embedding dimension compatible with the downstream LLM. The resulting structural token embeddings $Z_S$ encode both the structural relationships captured by the graph and the dynamic patterns captured by the structural-temporal embeddings.

**Temporal Tokenizer** $(T)$    To handle sequences of varying lengths (heterogeneity), we apply i) *Padding:* Sequences shorter than $n_{\max}$ are zero-padded to ensure uniform input dimensions and ii) *Masking:* Mask $M \in \mathbb{R}^{n_{\max}}$ indicates valid time steps, with marked real samples: 1 and padded samples: 0. This ensures the model focuses computations on valid temporal entries. The padded temporal sequence $\mathbf{X}_T^{\text{padded}}$ is combined with the structural-temporal embedding $\mathbf{E}(\mathbf{X}_i)$:

$$\mathbf{X}_T = \mathbf{X}_T^{\text{padded}} || \mathbf{E}(\mathbf{X}_i), \quad \mathbf{X}_T \in \mathbb{R}^{(n_{\max}+p+n_i) \times p} \tag{2}$$

where $n_{\max}$ is the maximum sequence length of longitudinal profile in the dataset. The concatenated temporal input is passed through a two-layer MLP. The first layer applies a nonlinear transformation $H_T = \text{ReLU}(X_T W_{T_1} + b_{T_1})$, $H_T \in \mathbb{R}^{(n_{\max}+p+n_i) \times d_{\text{hidden}}}$, followed by a second linear layer $Z_T^{\text{MLP}} = H_T W_{T_2} + b_{T_2}$, $Z_T^{\text{MLP}} \in \mathbb{R}^{(n_{\max}+p+n_i) \times d_{LLM}}$. While the MLP processes the entire longitudinal profile, the mask $M$ ensures only valid time steps influence the learned embeddings. Finally, layer normalization is applied to stabilize learning and ensure consistent scaling $Z_T = \text{LayerNorm}(Z_T^{\text{MLP}})$, $Z_T \in \mathbb{R}^{(n_{\max}+p+n_i) \times d_{LLM}}$. The resulting temporal token embeddings $Z_T$ effectively capture both the temporal evolution while preserving structural context.

## 3.3 Model Training

The output token embeddings $Z_{\text{Pre}}$, $Z_S$, and $Z_T$ are concatenated $\mathbf{Z} = Z_{\text{Pre}} || Z_S || Z_T$, where $\mathbf{Z} \in \mathbb{R}^{L \times d_{LLM}}$, with $L$ denoting the token sequence length and $d_{LLM}$ the embedding dimension compatible with the LLM backbone. This combined representation is passed to a pre-trained LLM, which has been augmented with LoRA adapter $\mathbf{O} = \text{Adapter}(\text{LLM})$, where $\mathbf{O}$ represents the model output for different downstream tasks, such as sequence prediction and anomaly detection over longitudinal profiles. During training, the tokenizers $(S, T)$ are trained jointly with the LoRA adapter, while the core LLM weights remain frozen. This setup allows efficient adaptation to specific downstream tasks with minimal computational overhead, leveraging the generalization capabilities of the pre-trained LLM while enabling domain-specific adaptation through the tokenizers and LoRA layers. The training objective functions can be defined according to the downstream task.

## 4 Experiments

We evaluate STT-LLM in the context of doping analytics in sports, where detecting abnormal steroid patterns over time is important for identifying potential prohibited drug abuse by athletes.

**Datasets**    The models are evaluated on a wide range of real-world athlete datasets (Table 1) consisting of longitudinal steroid profiles from their urine samples: Steroid-M (male), Steroid-F (female), Steroid-M$_{\text{lim}}$ (male, limited), and Steroid-F$_{\text{lim}}$ (female, limited) (Rahman et al., 2022). The dataset includes measurements of six key steroid metabolites: testosterone (T),

Table 1: Summary statistics of all the datasets.

| Datasets | Gender | # Profiles | # Samples | Length $n_i$ |
|---|---|---|---|---|
| Steroid-M | Male | 755 | 4214 | 3-20 |
| Steroid-F | Female | 375 | 2307 | 3-20 |
| Steroid-M$_{\text{lim}}$ | Male | 737 | 1474 | 2 |
| Steroid-F$_{\text{lim}}$ | Female | 293 | 586 | 2 |

epitestosterone (E), etiocholanolone (Etio), androsterone (A), $5\alpha$-androstanediol ($5\alpha$Adiol), and $5\beta$-androstanediol ($5\beta$Adiol) following the steroid metabolism pathway to synthesize (Piper et al., 2021). The profile lengths range from 2-20 samples per athlete, reflecting realistic variability in longitudinal monitoring. These datasets cover diverse population groups and temporal resolutions, allowing us to comprehensively evaluate STT-LLM under realistic conditions.

**Baselines** We compare the STT-LLM tokenization approach against different general-purpose mid-sized LLMs, including Qwen-2.5 (7B) (Yang et al., 2025), Falcon-3 (7B) (Almazrouei et al., 2023), Mistral (7B) (Jiang et al., 2023), LLaMA-2 (7B) (Touvron et al., 2023), LLaMA-3.1 (8B) (Grattafiori et al., 2024), Phi-4 (7B) (Abdin et al., 2024), and DeepSeek-R1 (7B) (DeepSeek-AI et al., 2025). Each model is fine-tuned on different downstream tasks using its native tokenization strategy. These models typically fall within the 7-8 billion parameter range, making them well-suited for efficient inference on local workstations without requiring large-scale GPU infrastructure.

**Experimental Setup** All experiments are conducted on a workstation equipped with an NVIDIA Titan RTX GPU (24GB), Intel i9 processor, and 31GB total RAM. We used the same computational setup for both STT-LLM and all baseline models to ensure fair and consistent comparisons. The evaluation was performed under two settings: *zero-shot*, and *few-shot* (2-20 labeled examples as in-context prompts). The evaluation metrics used are RMSE, MAE, and MAPE for sequence prediction, and accuracy, sensitivity, precision, F1-score, and AUC for anomaly detection. We set the high specificity value (99.9%) to avoid any false positives (domain requirements). All reported results are averaged over three independent runs with standard deviations reported where applicable.

## 5 RESULTS

### 5.1 SEQUENCE PREDICTION

**Zero-shot setting** Fig. 2 shows that STT-LLM consistently outperforms all LLM baselines by achieving the lowest error scores. For Steroid-M and Steroid-F, STT-LLM reduces RMSE value (%100) to 79.3 and 68.4, respectively, while all baselines remain above 83, indicating its improved ability to model metabolic patterns even without supervision. The gains are even more pronounced in the limited datasets, where STT-LLM achieves low RMSE value (%100) of 30.0 and 1.2, respectively, outperforming the next-best models by large margins. For MAE value ((%10)), STT-LLM consistently achieves the lowest errors across datasets, with values dropping to near 5-6 on the limited datasets, reflecting accurate point-wise predictions.

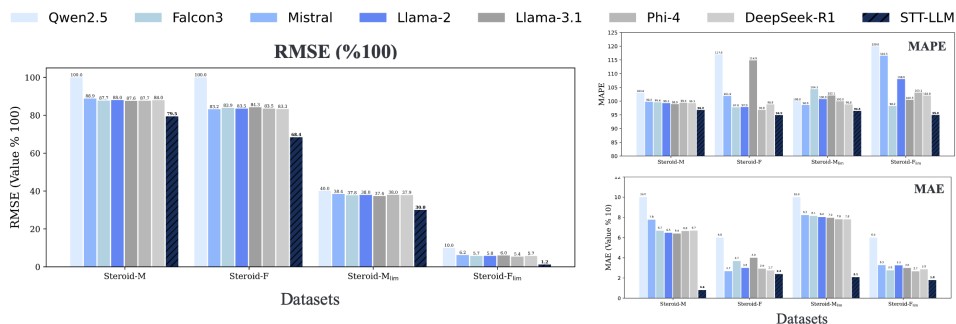

Figure 2: Zero-shot sequence prediction performance across different datasets.

**Few-shot setting** Table 7 shows several important findings. Contrary to expectations, the error metrics increase as the number of shots increases from 5 to 20 across all the models. This indicates that simply increasing the number of in-context examples does not necessarily improve performance. The rise in error is likely due to including heterogeneous and potentially noisy profiles as prompts, which may confuse the model instead of guiding it, especially in a domain like longitudinal clinical monitoring, where inter-individual variation is high. Despite this, STT-LLM consistently achieves the best RMSE across all datasets and shot counts, demonstrating robust temporal generalization. For example, at 5-shot, STT-LLM achieves the lowest RMSE on Steroid-$M_{lim}$ (1730.11), Steroid-$F_{lim}$ (1276.32), and maintains higher performance across more shots as well. Similarly, in terms of MAE, our model outperforms baselines on Steroid-$F_{lim}$ with a score of 643.71 (10-shot) and 642.90 (20-shot). STT-LLM maintains high overall stability and minimal fluctuation in MAPE compared to LLM baselines. These findings suggest that STT-LLM outperforms baselines consistently in absolute error terms and demonstrates better resilience to prompt variability and shot-induced drift.

Table 2: Few-shot sequence prediction results across different datasets.

| Datasets | Model | @5 | | | @10 | | | @15 | | | @20 | | |
|---|---|---|---|---|---|---|---|---|---|---|---|---|---|
| | | RMSE↓ | MAE↓ | MAPE↓ | RMSE↓ | MAE↓ | MAPE↓ | RMSE↓ | MAE↓ | MAPE↓ | RMSE↓ | MAE↓ | MAPE↓ |
| Steroid-M | Qwen-2.5 | 1695.99 | 899.99 | 111.99 | 1695.99 | 899.99 | 111.99 | 1695.99 | 899.99 | 111.99 | 1695.99 | 899.99 | 111.99 |
| | Mistral | 1688.34 | 894.92 | 98.19 | 1690.63 | 896.48 | **94.69** | 1688.90 | 896.54 | 101.04 | 1692.39 | 899.84 | 110.19 |
| | Falcon-3 | 1688.02 | 896.54 | 101.17 | 1689.88 | 897.20 | 100.31 | 1690.39 | 897.48 | 100.01 | 1691.48 | 897.69 | 100.93 |
| | LLaMA-2 | 1687.80 | 895.81 | 98.21 | 1689.47 | 897.29 | 100.74 | 1690.01 | 896.86 | 100.47 | 1691.16 | 897.06 | 98.19 |
| | LLaMA-3.1 | 1688.57 | 896.78 | 100.27 | 1689.67 | 898.19 | 106.75 | 1690.56 | 897.17 | 101.46 | 1690.98 | 896.84 | 97.21 |
| | Phi-4 | 1688.20 | 896.62 | 100.38 | 1690.17 | 897.21 | 97.98 | 1690.04 | 897.36 | 102.87 | 1691.41 | 897.54 | 100.88 |
| | DeepSeek-R1 | 1688.05 | 896.73 | 100.33 | 1689.88 | 896.73 | 98.31 | 1690.31 | 897.17 | 98.81 | 1691.65 | 897.56 | 99.48 |
| | **STT-LLM** | **1680.00** | **890.77** | **96.80** | **1681.57** | **891.27** | 96.79 | **1682.06** | **891.37** | 96.79 | **1683.51** | **891.87** | **96.81** |
| Steroid-F | Qwen-2.5 | 1395.99 | 699.99 | 129.99 | 1395.99 | 699.99 | 129.99 | 1395.99 | 699.99 | 129.99 | 1395.99 | 699.99 | 129.99 |
| | Mistral | 1387.98 | 695.23 | 120.35 | 1392.05 | 697.48 | **92.08** | 1388.75 | 695.68 | 108.68 | 1390.98 | 696.63 | 107.37 |
| | Falcon-3 | 1388.12 | 695.07 | 93.71 | 1389.62 | 695.41 | 93.80 | 1388.77 | 695.67 | 115.24 | 1389.53 | 694.61 | **94.31** |
| | LLaMA-2 | 1387.93 | 694.93 | **93.30** | 1388.86 | 695.52 | 109.01 | 1388.92 | 694.91 | 100.61 | 1389.93 | 695.33 | 101.76 |
| | LLaMA-3.1 | 1388.67 | 695.34 | 94.12 | 1389.38 | 695.03 | 93.93 | 1388.72 | 695.19 | 106.55 | 1390.03 | 695.23 | 103.50 |
| | Phi-4 | 1388.07 | 695.39 | 98.83 | 1389.09 | 695.64 | 108.44 | 1389.50 | 694.70 | 99.72 | 1389.75 | 695.43 | 108.37 |
| | DeepSeek-R1 | 1388.48 | 695.47 | 102.55 | 1389.54 | 696.04 | 97.25 | 1389.05 | 695.14 | 98.93 | 1389.09 | 694.41 | 95.36 |
| | **STT-LLM** | **1372.85** | **684.39** | 94.94 | **1374.17** | **684.89** | 94.92 | **1373.51** | **684.05** | 94.91 | **1374.45** | **684.09** | 94.91 |
| Steroid-M$_{lim}$ | Qwen-2.5 | 1750.99 | 901.99 | 106.99 | 1750.99 | 901.99 | 106.99 | 1750.99 | 901.99 | 106.99 | 1750.99 | 901.99 | 106.99 |
| | Mistral | 1737.63 | 896.17 | **95.80** | 1742.02 | 899.45 | 103.07 | 1738.92 | 898.42 | 103.42 | 1741.69 | 898.80 | 98.51 |
| | Falcon-3 | 1738.66 | 898.03 | 102.52 | 1740.75 | 898.42 | 98.74 | 1738.93 | 898.31 | 102.14 | 1742.69 | 900.19 | 97.78 |
| | LLaMA-2 | 1738.65 | 897.73 | 99.46 | 1741.24 | 898.86 | 100.51 | 1738.90 | 898.48 | 103.02 | 1743.15 | 900.89 | 102.91 |
| | LLaMA-3.1 | 1737.29 | 896.35 | 100.98 | 1741.25 | 899.54 | 100.01 | 1739.00 | 898.11 | 98.56 | 1743.21 | 900.77 | 98.70 |
| | Phi-4 | 1738.51 | 898.01 | 102.96 | 1741.42 | 898.43 | 97.71 | 1738.87 | 898.07 | 99.79 | 1743.05 | 900.66 | 98.94 |
| | DeepSeek-R1 | 1738.12 | 897.42 | 100.40 | 1740.81 | 898.72 | 98.67 | 1739.72 | 898.67 | 99.97 | 1743.62 | 900.59 | 98.15 |
| | **STT-LLM** | **1730.11** | **891.67** | 96.47 | **1733.18** | **893.01** | **96.47** | **1731.43** | **892.63** | **96.47** | **1734.87** | **894.61** | **96.47** |
| Steroid-F$_{lim}$ | Qwen-2.5 | 1309.99 | 666.99 | 127.99 | 1309.99 | 666.99 | 127.99 | 1309.99 | 666.99 | 127.99 | 1309.99 | 666.99 | 127.99 |
| | Mistral | 1292.73 | 657.49 | 123.29 | 1289.67 | 654.38 | 118.12 | 1294.05 | 657.05 | 107.06 | 1286.36 | 652.15 | 126.36 |
| | Falcon-3 | 1291.65 | 655.82 | 97.87 | 1289.63 | 654.51 | 96.85 | 1294.77 | 656.69 | 102.03 | 1287.89 | 653.04 | 96.47 |
| | LLaMA-2 | 1292.06 | 656.18 | 100.69 | 1289.05 | 653.63 | 93.96 | 1295.08 | 656.57 | 101.13 | 1287.68 | 654.31 | 106.19 |
| | LLaMA-3.1 | 1291.13 | 654.67 | **88.66** | 1289.66 | 654.13 | **93.20** | 1293.98 | 656.87 | 115.68 | 1287.32 | 653.76 | 108.22 |
| | Phi-4 | 1291.92 | 655.84 | 92.49 | 1289.48 | 654.17 | 97.54 | 1294.68 | 656.39 | 102.23 | 1287.37 | 653.85 | 107.41 |
| | DeepSeek-R1 | 1291.64 | 655.94 | 101.25 | 1289.33 | 654.37 | 103.85 | 1294.89 | 656.59 | 97.21 | 1286.90 | 653.18 | 101.71 |
| | **STT-LLM** | **1276.32** | **645.16** | 94.92 | **1274.23** | **643.71** | 94.89 | **1279.59** | **645.90** | 94.89 | **1272.19** | **642.90** | **94.86** |

## 5.2 ANOMALY DETECTION

**Zero-shot setting** Table 3 shows that STT-LLM significantly outperforms baseline models in both local and global anomaly detection under zero-shot conditions. For local anomaly detection, STT-LLM achieves sensitivity of 15.0% on Steroid-M and 17.0% on Steroid-F$_{lim}$, while most baselines show near-zero sensitivity. This is because these models default to classifying all samples as normal, resulting in artificially inflated accuracy values around 95-96% but completely failing to identify any anomalous samples. In contrast, STT-LLM trades a small drop in accuracy (87-88%) for substantial gains in sensitivity and precision, reflecting its ability to detect true anomalies. In global anomaly detection, all models achieve better accuracy, as the classification task is inherently less sparse and the signal-to-noise ratio is higher. STT-LLM achieves the highest F1-scores (0.26 on Steroid-M, 0.29 on Steroid-F) and AUC values (0.57 on Steroid-M, 0.59 on Steroid-F), outperforming baselines by up to ∼10%. These results highlight STT-LLM's ability to handle both sparse (local) and dense (global) anomaly tasks, demonstrating increased robustness and generalization compared to standard LLMs, especially in rare-event detection scenarios where sensitivity is critical.

**Few-shot setting** Fig. 3 shows that STT-LLM achieves substantial gains in global anomaly detection as the number of shots increases. Unlike the baselines, which often exhibit unstable or noisy trends across shot sizes, STT-LLM shows consistent improvements across most metrics. For sensitivity, STT-LLM increases from 0.15 (2 shots) to 0.6 (20 shots) on Steroid-M, representing more than a threefold improvement in detecting true anomalies. Precision improves steadily as well, reaching near-perfect levels on Steroid-F and Steroid-F$_{lim}$, indicating that the model sharply reduces false positives as supervision increases. F1-score trends further highlight the balanced gains of STT-LLM, with performance rising sharply between 2 and 20 shots, e.g., 0.15 to 0.7 (Steroid-M), demonstrating the model's ability to jointly improve sensitivity and precision. Overall, STT-LLM's performance curves remain smooth, while baselines frequently show oscillating or deteriorating patterns as shots increase, reflecting their difficulty in integrating few-shot supervision effectively.

## 5.3 ABLATION STUDY

The ablations include removing all components (*w/o all*), structural tokenizer (*w/o structural*), temporal tokenizer (*w/o temporal*), embedding layer (*w/o embeddings*), and pairs of components.

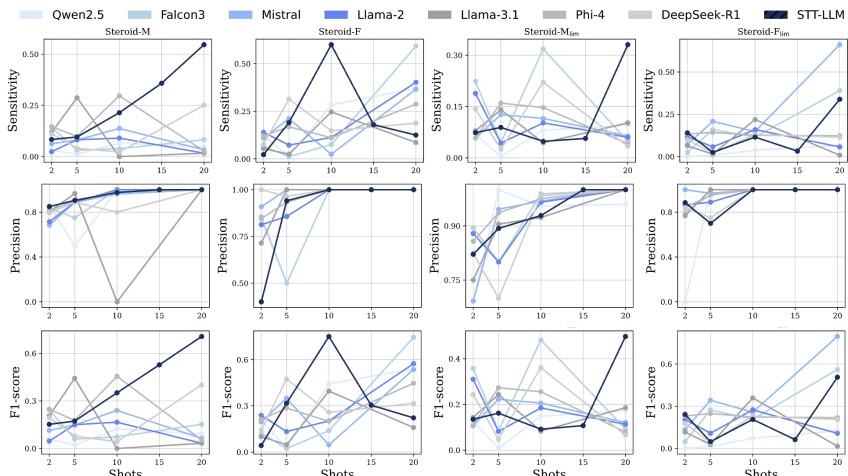

Figure 3: Few-shot global anomaly detection performance across different metrics.

Table 3: Local and global anomaly detection results across different datasets at zero-shot setting.

| Datasets | Model | Local | | | | | Global | | | | |
|---|---|---|---|---|---|---|---|---|---|---|---|
| | | Acc↑ | Sens↑ | Prec↑ | F1↑ | AUC↑ | Acc↑ | Sens↑ | Prec↑ | F1↑ | AUC↑ |
| Steroid-M | Qwen-2.5 | **0.96**±.01 | 0.00±.00 | 0.00±.00 | 0.00±.00 | 0.47±.02 | 0.71±.02 | 0.08±.03 | 0.20±.02 | 0.11±.02 | 0.45±.02 |
| | Mistral | 0.87±.02 | 0.05±.01 | 0.02±.01 | 0.03±.01 | 0.43±.02 | 0.71±.02 | 0.08±.02 | 0.23±.03 | 0.12±.02 | 0.47±.02 |
| | Falcon-3 | 0.94±.02 | 0.01±.00 | 0.02±.01 | 0.01±.01 | 0.46±.02 | 0.72±.02 | 0.08±.02 | 0.28±.03 | 0.13±.02 | 0.53±.02 |
| | LLaMA-2 | 0.90±.02 | 0.05±.01 | 0.03±.01 | 0.04±.01 | 0.42±.02 | 0.71±.02 | 0.09±.02 | 0.26±.02 | 0.14±.03 | 0.49±.02 |
| | LLaMA-3.1 | 0.87±.02 | 0.07±.01 | 0.03±.01 | 0.05±.01 | 0.51±.02 | 0.72±.02 | 0.14±.02 | 0.33±.03 | 0.19±.03 | 0.56±.02 |
| | Phi-4 | 0.87±.02 | 0.08±.01 | 0.04±.01 | 0.05±.01 | 0.50±.02 | 0.72±.02 | 0.03±.01 | 0.15±.02 | 0.05±.01 | 0.46±.02 |
| | DeepSeek-R1 | 0.95±.01 | 0.02±.01 | 0.01±.01 | 0.01±.00 | 0.39±.02 | 0.70±.02 | 0.08±.02 | 0.21±.02 | 0.11±.02 | 0.45±.02 |
| | **STT-LLM** | 0.87±.02 | **0.15**±.02 | **0.07**±.01 | **0.09**±.02 | **0.57**±.02 | **0.73**±.02 | **0.19**±.03 | **0.41**±.03 | **0.26**±.03 | **0.57**±.02 |
| Steroid-F | Qwen-2.5 | 0.87±.02 | 0.04±.01 | 0.02±.01 | 0.02±.01 | 0.46±.02 | 0.73±.02 | 0.04±.01 | 0.14±.02 | 0.06±.01 | 0.55±.02 |
| | Mistral | **0.96**±.01 | 0.00±.00 | 0.00±.00 | 0.00±.00 | 0.62±.02 | 0.73±.02 | 0.12±.02 | 0.26±.03 | 0.16±.02 | 0.43±.02 |
| | Falcon-3 | 0.95±.01 | 0.00±.00 | 0.00±.00 | 0.00±.00 | 0.60±.02 | 0.72±.02 | 0.09±.02 | 0.22±.02 | 0.13±.02 | 0.37±.02 |
| | LLaMA-2 | 0.87±.02 | 0.06±.01 | 0.02±.01 | 0.03±.01 | 0.55±.02 | 0.73±.02 | 0.12±.02 | 0.26±.02 | 0.16±.02 | 0.47±.02 |
| | LLaMA-3.1 | 0.95±.01 | 0.01±.00 | 0.03±.01 | 0.02±.01 | 0.57±.02 | 0.73±.02 | 0.10±.02 | 0.23±.03 | 0.14±.02 | 0.49±.02 |
| | Phi-4 | 0.88±.02 | 0.06±.01 | 0.02±.01 | 0.03±.01 | 0.50±.02 | 0.74±.02 | 0.08±.02 | 0.25±.02 | 0.13±.02 | 0.45±.02 |
| | DeepSeek-R1 | 0.87±.02 | 0.06±.01 | 0.02±.01 | 0.03±.01 | 0.42±.02 | 0.73±.02 | 0.10±.02 | 0.22±.03 | 0.13±.02 | 0.50±.02 |
| | **STT-LLM** | 0.87±.02 | **0.08**±.01 | **0.03**±.01 | **0.05**±.01 | 0.47±.02 | **0.75**±.02 | **0.23**±.03 | **0.40**±.03 | **0.29**±.03 | **0.59**±.02 |
| Steroid-M$_{lim}$ | Qwen-2.5 | 0.86±.02 | 0.00±.00 | 0.00±.00 | 0.00±.00 | 0.18±.01 | 0.62±.02 | 0.06±.02 | 0.30±.03 | 0.10±.02 | 0.54±.02 |
| | Mistral | **0.96**±.01 | 0.00±.00 | 0.00±.00 | 0.00±.00 | 0.37±.02 | 0.61±.02 | 0.07±.02 | 0.31±.02 | 0.12±.02 | 0.42±.02 |
| | Falcon-3 | 0.88±.02 | 0.08±.01 | 0.03±.01 | 0.05±.01 | 0.44±.02 | 0.61±.02 | 0.07±.02 | 0.32±.02 | 0.12±.02 | 0.53±.02 |
| | LLaMA-2 | 0.88±.02 | 0.03±.01 | 0.01±.01 | 0.02±.01 | 0.22±.01 | 0.61±.02 | 0.07±.02 | 0.31±.02 | 0.12±.02 | 0.45±.02 |
| | LLaMA-3.1 | 0.88±.02 | 0.04±.01 | 0.01±.01 | 0.02±.01 | 0.39±.02 | 0.60±.02 | 0.04±.01 | 0.21±.02 | 0.07±.02 | 0.39±.02 |
| | Phi-4 | 0.89±.02 | 0.21±.02 | 0.09±.01 | 0.12±.02 | 0.65±.02 | 0.61±.02 | 0.05±.01 | 0.25±.02 | 0.09±.01 | 0.44±.02 |
| | DeepSeek-R1 | 0.87±.02 | 0.06±.01 | 0.02±.01 | 0.03±.01 | 0.43±.02 | 0.60±.02 | 0.04±.01 | 0.19±.02 | 0.07±.01 | 0.45±.02 |
| | **STT-LLM** | 0.88±.02 | **0.36**±.02 | **0.12**±.02 | **0.18**±.02 | **0.75**±.02 | **0.64**±.02 | **0.12**±.02 | **0.47**±.03 | **0.19**±.02 | **0.55**±.02 |
| Steroid-F$_{lim}$ | Qwen-2.5 | 0.88±.02 | 0.06±.01 | 0.06±.01 | 0.06±.01 | 0.14±.01 | 0.54±.02 | 0.10±.02 | 0.46±.03 | 0.16±.02 | 0.53±.02 |
| | Mistral | 0.95±.01 | 0.01±.00 | 0.03±.00 | 0.02±.00 | **0.64**±.02 | 0.51±.02 | 0.04±.01 | 0.25±.02 | 0.07±.01 | 0.42±.02 |
| | Falcon-3 | **0.96**±.01 | 0.00±.00 | 0.00±.00 | 0.00±.00 | 0.27±.02 | 0.55±.02 | 0.13±.02 | 0.53±.03 | 0.21±.03 | 0.55±.02 |
| | LLaMA-2 | 0.86±.02 | 0.03±.01 | 0.01±.01 | 0.02±.01 | 0.32±.02 | 0.54±.02 | 0.11±.02 | 0.48±.03 | 0.18±.02 | 0.50±.02 |
| | LLaMA-3.1 | 0.87±.02 | 0.00±.00 | 0.00±.00 | 0.00±.00 | 0.08±.01 | 0.52±.02 | 0.07±.01 | 0.36±.02 | 0.11±.01 | 0.46±.02 |
| | Phi-4 | 0.87±.02 | 0.07±.01 | 0.03±.01 | 0.04±.01 | 0.48±.02 | 0.53±.02 | 0.07±.01 | 0.41±.03 | 0.12±.02 | 0.48±.02 |
| | DeepSeek-R1 | 0.86±.02 | 0.10±.01 | 0.04±.01 | 0.06±.01 | 0.51±.02 | 0.54±.02 | 0.10±.02 | 0.48±.03 | 0.16±.02 | 0.54±.02 |
| | **STT-LLM** | 0.87±.02 | **0.17**±.02 | **0.08**±.01 | **0.11**±.02 | 0.54±.02 | **0.59**±.02 | **0.15**±.03 | **0.71**±.03 | **0.25**±.03 | **0.56**±.02 |

Table 4 shows that STT-LLM achieves the lowest sequence prediction errors (RMSE: 1664.59, MAPE: 96.80). Removing all components increases RMSE: +1.4%, MAE: +1.7%, MAPE: +2.2% relative to STT-LLM. Removing embeddings alone increases MAPE to 100.56 (+3.9%)

Table 4: Contributions of different components in STT-LLM.

| Model Variants | Sequence Prediction | | | Anomaly Detection (Global) | | | | |
|---|---|---|---|---|---|---|---|---|
| | RMSE↓ | MAE↓ | MAPE↓ | Acc↑ | Sens↑ | Prec↑ | F1↑ | AUC↑ |
| *w/o* all | 1687.71 | 896.39 | 98.93 | 0.7179 | 0.1398 | 0.3291 | 0.1962 | 0.5609 |
| *w/o* structural | 1687.49 | 896.61 | 100.65 | 0.7152 | 0.0968 | 0.2769 | 0.1434 | 0.4964 |
| *w/o* temporal | 1682.45 | 892.85 | 98.38 | 0.7126 | 0.1237 | 0.2987 | 0.1749 | 0.5500 |
| *w/o* embeddings | 1682.75 | 893.40 | 100.56 | 0.7139 | 0.1344 | 0.3125 | 0.1880 | 0.5352 |
| *w/o* structural + temporal | 1682.70 | 893.20 | 98.89 | 0.6967 | 0.0645 | 0.1791 | 0.0949 | 0.4877 |
| *w/o* embeddings + temporal | 1677.56 | 889.29 | 97.07 | 0.7245 | 0.1290 | 0.3429 | 0.1875 | 0.5474 |
| *w/o* embeddings + structural | 1679.16 | 891.78 | 97.35 | 0.7113 | 0.0914 | 0.2576 | 0.1349 | 0.4887 |
| **STT-LLM** | **1664.59** | **881.20** | **96.80** | **0.7338** | **0.1935** | **0.4138** | **0.2637** | **0.5675** |

and drops AUC to 0.5352 (-5.7%), highlighting the embedding layer's key role in aligning multimodal representations. For anomaly detection, STT-LLM achieves a good balance across different metrics. Removing all components lowers sensitivity by -27.8%, and precision by -20.5%

compared to STT-LLM. Removing either the structural or temporal tokenizer reduces sensitivity by -50% (0.0968 - 0.1237) and precision by -33% (0.2769 - 0.2987), showing that both structural and temporal components are important for anomaly detection. When two components are removed, the degradation is even sharper, e.g., *w/o embeddings + structural* drops AUC by -14% (0.4887) relative to STT-LLM. The *w/o all* variant performs slightly better than some partial ablations because complete removal avoids embedding mismatches and produces uniform flat inputs, whereas partial removal yields incoherent fused representations that confuse the model.

# 6    CASE STUDY

To evaluate the real-world applicability of our method, we conducted a case study on 29 longitudinal steroid profiles from real-world athletes, which were verified through DNA analysis by an anti-doping laboratory. Among these, 7 profiles were confirmed as anomalous due to different doping-related abnormalities, with domain experts providing detailed explanations, and the remaining 22 were classified as clean profiles. We used the clean profiles for sequence prediction and all 29 for anomaly detection. Our model achieved better forecasting performance with RMSE: 1673.13, MAE: 868.93, and MAPE: 95.51. For anomaly detection, the model accurately identified all 7 anomalous cases with 100% sensitivity, while misclassifying only 2 clean profiles (accuracy: 93.10%).

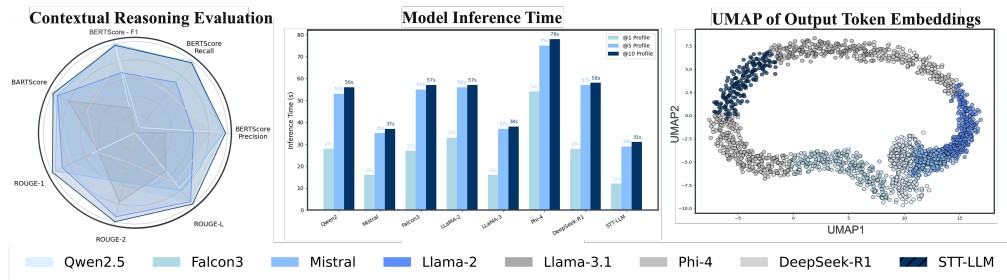

Figure 4: Evaluation of contextual reasoning quality (left), model inference time (center), and combined UMAP projection of output token embeddings from STT-LLM and LLM baselines (right).

To evaluate the contextual reasoning ability of STT-LLM, we adopt a few-shot setup using the 7 expert-annotated longitudinal profiles to generate explanations for 500 additional profiles. These explanations were used to train all the models under identical training conditions. We then assessed model performance on the original 7 profiles (expert ground-truth explanations). As shown in Figure 4, STT-LLM outperforms all competing LLM baselines across multiple evaluation metrics, showing higher alignment with expert interpretations. This highlights the model's ability to capture clinically meaningful reasoning patterns from limited supervision. In addition, we compared the inference efficiency of all models across different profile settings (1, 5, and 10 profiles). STT-LLM achieved substantially lower inference times than the baselines, requiring only 12s, 29s, and 31s respectively, whereas alternative LLMs ranged between 27-78s depending on model size and profile count. Finally, we visualize the output token embedding spaces of different models using UMAP representation. Unlike tightly clustered distributions, the embeddings form a continuous ring-like topology, suggesting a shared latent manifold, where STT-LLM occupies a transitional zone between LLaMA-3.1 and Phi-4. This placement suggests that STT-LLM maintains representational alignment with general-purpose LLMs while introducing localized structure unique to its domain-aware training.

# 7    CONCLUSION

We introduce the STT-LLM framework, which enables LLMs to analyze longitudinal biomedical profiles. By constructing joint embeddings and applying specialized tokenization strategy to capture both temporal dynamics and domain-specific structural dependencies, STT-LLM allows LLMs to adapt to different longitudinal clinical tasks. While we acknowledge that graph-based models are strong baselines for modeling structural dependencies, they were not included in our comparison since their architectures are not directly compatible with LLM inference pipelines. Future studies could incorporate such domain-specific models to provide an even broader perspective on the trade-offs.

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

# A  TECHNICAL APPENDIX

## A.1  EXPERIMENTAL SETUP AND MODEL HYPERPARAMETERS

**Model Configuration**   STT-LLM was trained using parameter-efficient fine-tuning via LoRA, where we systematically evaluated the impact of key hyperparameters on model performance. The final configuration uses a LoRA rank of 32, scaling factor (alpha) of 128, and a learning rate of 2e-5. To assess the sensitivity of the model to these choices, we conducted experiments on the Steroid-M dataset by varying one hyperparameter at a time while keeping others fixed. As shown in Table 5, reducing the LoRA rank to 16 led to a slight degradation in both sequence prediction (RMSE↑ +18.6) and anomaly detection (AUC↓ -0.006), while increasing the rank to 64 did not yield further gains. Similarly, modifying the alpha parameter to 64 or 256 degraded both predictive accuracy and detection precision, suggesting that 128 offers a balanced regularization. Finally, tuning the learning rate revealed that deviating from 2e-5, either lower (1e-6) or higher (2e-4) consistently reduced performance, particularly in terms of F1-score and AUC. These findings indicate that the selected configuration for STT-LLM strikes an optimal balance between predictive accuracy and detection sensitivity under constrained fine-tuning conditions. All experiments were conducted for 10 epochs using early stopping on a single NVIDIA Titan RTX GPU with 24GB memory.

**Projection Dimension and MLP Depth Sensitivity**   To assess the architectural design of our tokenizers, we study the impact of varying the projection dimension (PD) and the number of MLP layers in the structural and temporal tokenizers of STT-LLM. Our default configuration uses a projection dimension of 4096 and two MLP layers per tokenizer. As shown in Table 6, reducing the depth to a single MLP layer leads to a noticeable drop in detection performance, particularly sensitivity (-8.6%) and AUC (-0.046). Increasing the depth to three layers does not yield further gains, indicating that two layers strike a balance between expressivity and generalization. Similarly, projection dimensions of 1024 and 2048 underperform the 4096-dimensional variant, especially on precision and F1-score. The model variant with 4096 PD and 2-layer MLPs achieves the highest performance across all anomaly detection metrics (e.g., AUC: 0.5675, F1: 0.2637), highlighting the importance of sufficient projection capacity and moderate depth in capturing clinically relevant temporal and structural patterns.

**Prompt Design for Task-Specific Supervision**   To enable task-specific training and evaluation across all models, including STT-LLM, we designed structured natural language prompts tailored to three core objectives: reasoning, classification, and sequence prediction. As illustrated in Figure 5, the reasoning prompt asks the model to detect and explain abnormalities within a given steroid profile across multiple time points, encouraging contextual understanding and interpretability. The classification prompt explicitly instructs the model to either confirm the consistency of normal profiles or identify and explain the presence of an anomaly in the final sample. In contrast, the prediction prompt emphasizes learning temporal patterns, either under the assumption of normality or while acknowledging that the last sample deviates from the expected trend. These prompts allow all models to operate in a unified few-shot setting while supporting gradient-based fine-tuning or embedding-level supervision, depending on the architecture. They also ensure that the models are evaluated consistently across both descriptive and diagnostic clinical tasks.

## A.2  DETAILED RESULTS

### A.2.1  MODEL LOSS PERFORMANCE

We report the training and evaluation loss curves for both sequence prediction and anomaly detection tasks over 10 training epochs in Figure 6. For sequence prediction (left), the model shows rapid convergence, with the training loss decreasing sharply within the first 5 epochs and plateauing around 620. The evaluation loss follows a similar trend, stabilizing around epoch 5, indicating strong generalization without overfitting. For anomaly detection (right), both training and evaluation losses exhibit an even faster convergence, with steep declines in the first 3 epochs and near-flattening thereafter around a value of 0.2. This consistency between train and eval curves in both tasks demonstrates that the STT-LLM framework effectively learns stable representations with minimal overfitting, even in low-resource settings. The rapid convergence further underscores the efficiency of

Table 5: Contributions of different hyperparameter configurations in STT-LLM on Steroid-M dataset.

| Model Variants | Sequence Prediction | | | Anomaly Detection (Global) | | | | |
|---|---|---|---|---|---|---|---|---|
| | RMSE↓ | MAE↓ | MAPE↓ | Acc↑ | Sens↑ | Prec↑ | F1↑ | AUC↑ |
| Lower Rank (16) | 1683.22 | 893.40 | 97.96 | 0.7179 | 0.1398 | 0.3291 | 0.1962 | 0.5609 |
| Higher Rank (64) | 1687.49 | 896.61 | 100.65 | 0.7152 | 0.0968 | 0.2769 | 0.1434 | 0.4964 |
| Lower alpha (64) | 1668.54 | 883.90 | 100.00 | 0.7126 | 0.1237 | 0.2987 | 0.1749 | 0.5500 |
| Higher alpha (256) | 1682.75 | 893.40 | 100.56 | 0.7139 | 0.1344 | 0.3125 | 0.1880 | 0.5352 |
| Lower LR (1e-6) | 1679.46 | 890.80 | 98.64 | 0.7245 | 0.0645 | 0.2609 | 0.1034 | 0.5034 |
| Higher LR (2e-4) | 1689.05 | 897.55 | 100.23 | 0.7126 | 0.1183 | 0.2933 | 0.1686 | 0.5005 |
| **STT-LLM** | **1664.59** | **881.20** | **96.80** | **0.7338** | **0.1935** | **0.4138** | **0.2637** | **0.5675** |

Table 6: Contributions of different Projection Dimensions (PD) and MLP layers in STT-LLM on Steroid-M dataset.

| Model Variants | Anomaly Detection (Global) | | | | |
|---|---|---|---|---|---|
| | Acc↑ | Sens↑ | Prec↑ | F1↑ | AUC↑ |
| 1 MLP layer | 0.7060 | 0.1075 | 0.2632 | 0.1527 | 0.5214 |
| 3 MLP layers | 0.7086 | 0.1183 | 0.2821 | 0.1667 | 0.5103 |
| 1024 PD | 0.7232 | 0.0753 | 0.2745 | 0.1181 | 0.5026 |
| 2048 PD | 0.7285 | 0.1129 | 0.3443 | 0.1700 | 0.5490 |
| **STT-LLM** | **0.7338** | **0.1935** | **0.4138** | **0.2637** | **0.5675** |

the proposed structural-temporal tokenization strategy, which enables fast adaptation to downstream clinical tasks with minimal tuning.

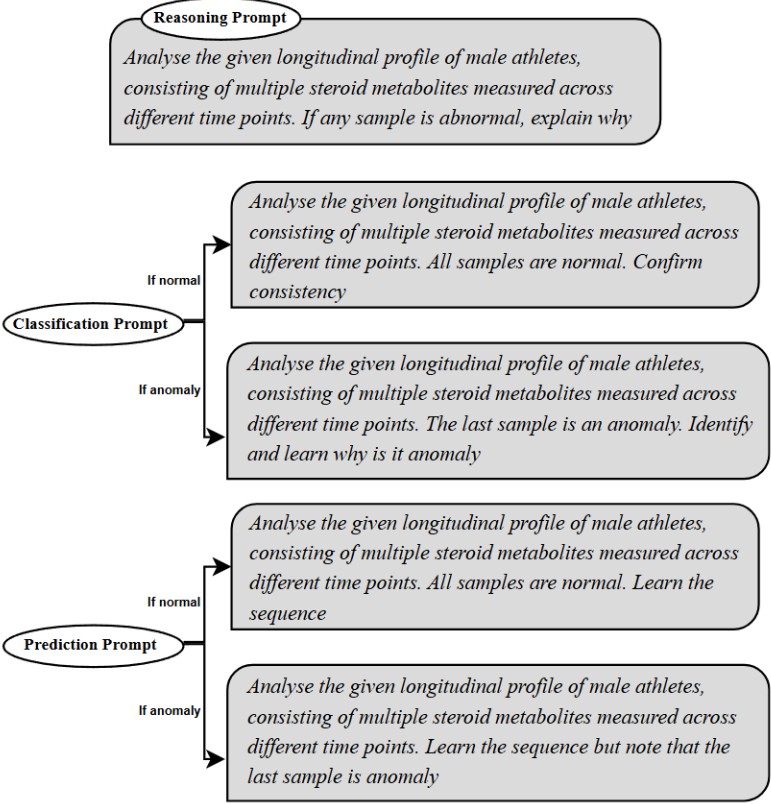

Figure 5: Prompts for training STT-LLM for different tasks.

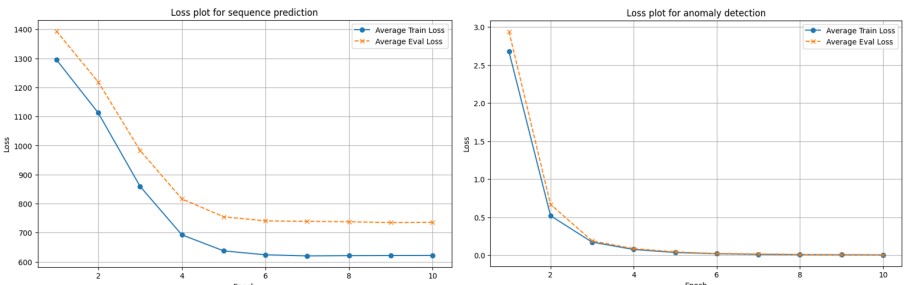

Figure 6: Loss plots for sequence prediction & anomaly detection tasks.

### A.2.2 SEQUENCE PREDICTION

**Few-shot settings** As shown in Figure 7 and Table 7, STT-LLM consistently outperforms all competing LLM baselines across all datasets and shot configurations in terms of RMSE, MAE, and MAPE. STT-LLM maintains stable and low error margins across shot variations, while other models (Qwen-2.5 and Mistral) show higher variance and sensitivity to shot count. On the Steroid-M dataset, STT-LLM achieves the best RMSE (1664.59), MAE (881.20), and MAPE (96.80) in the 2-shot setting and retains this advantage throughout. This performance trend generalizes to the limited-data settings (Steroid-$M_{lim}$, Steroid-$F_{lim}$), where the tokenization-aware STT-LLM demonstrates stronger robustness and lower generalization error. These results validate the model's ability to learn meaningful temporal patterns under constrained supervision and emphasize the effectiveness of its structural-temporal embedding design for few-shot sequence modeling in longitudinal clinical data.

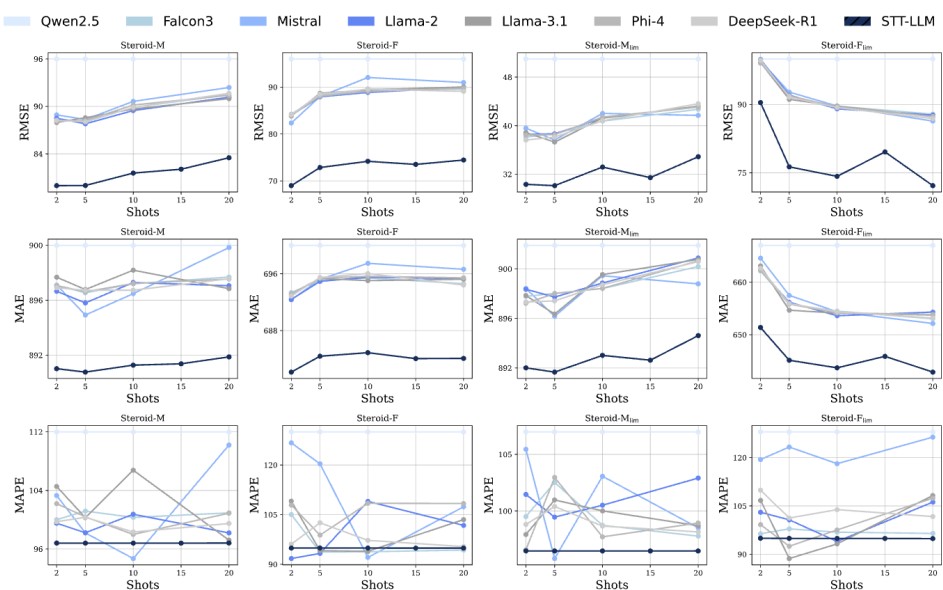

Figure 7: Few-shot learning results for sequence prediction.

### A.2.3 ANOMALY DETECTION

**Zero-Shot Local Anomaly Detection** Figure 8 presents the zero-shot local anomaly detection performance across four datasets. STT-LLM consistently outperforms all baseline models across all metrics. It achieves the highest sensitivity on Steroid-M and Steroid-$M_{lim}$ (16.8% and 32.4%, respectively), which are particularly challenging due to subtle temporal deviations. Additionally, STT-LLM shows significantly higher F1-scores (up to 18.4%) and precision values compared to all baselines, indicating its capacity to correctly identify rare anomalous samples with minimal

Table 7: Few-shot (2, 5, 10, 15, 20) sequence prediction results across different datasets.

| Datasets | Model | @2 | | | @5 | | | @10 | | | @15 | | | @20 | | |
|---|---|---|---|---|---|---|---|---|---|---|---|---|---|---|---|---|
| | | RMSE↓ | MAE↓ | MAPE↓ | RMSE↓ | MAE↓ | MAPE↓ | RMSE↓ | MAE↓ | MAPE↓ | RMSE↓ | MAE↓ | MAPE↓ | RMSE↓ | MAE↓ | MAPE↓ |
| Steroid-M | Qwen-2.5 | 1695.99 | 899.99 | 111.99 | 1695.99 | 899.99 | 111.99 | 1695.99 | 899.99 | 111.99 | 1695.99 | 899.99 | 111.99 | 1695.99 | 899.99 | 111.99 |
| | Mistral | 1688.93 | 897.11 | 103.29 | 1688.34 | 894.92 | 98.19 | 1690.63 | 896.48 | 94.69 | 1688.90 | 896.54 | 101.04 | 1692.39 | 899.84 | 110.19 |
| | Falcon-3 | 1688.43 | 897.08 | 100.00 | 1688.02 | 896.54 | 101.17 | 1689.88 | 897.20 | 100.31 | 1690.39 | 897.48 | 100.01 | 1691.48 | 897.69 | 100.93 |
| | LLaMA-2 | 1688.50 | 896.65 | 99.51 | 1687.80 | 895.81 | 98.21 | 1689.47 | 897.29 | 100.74 | 1690.01 | 896.86 | 100.47 | 1691.16 | 897.06 | 98.19 |
| | LLaMA-3.1 | 1687.96 | 897.67 | 104.53 | 1688.57 | 896.78 | 100.27 | 1689.67 | 898.19 | 106.75 | 1690.56 | 897.17 | 101.46 | 1690.98 | 896.84 | 97.21 |
| | Phi-4 | 1688.14 | 897.07 | 102.19 | 1688.20 | 896.62 | 100.38 | 1690.17 | 897.21 | 97.98 | 1690.04 | 897.36 | 102.87 | 1691.41 | 897.54 | 100.88 |
| | DeepSeek-R1 | 1688.25 | 896.91 | 99.72 | 1688.05 | 896.73 | 100.33 | 1689.88 | 896.73 | 98.31 | 1690.31 | 897.17 | 98.81 | 1691.65 | 897.56 | 99.48 |
| | **STT-LLM** | **1679.99** | **891.02** | **96.80** | **1680.00** | **890.77** | **96.80** | **1681.57** | **891.27** | **96.79** | **1682.06** | **891.37** | **96.79** | **1683.51** | **891.87** | **96.81** |
| Steroid-F | Qwen-2.5 | 1395.99 | 699.99 | 129.99 | 1395.99 | 699.99 | 129.99 | 1395.99 | 699.99 | 129.99 | 1395.99 | 699.99 | 129.99 | 1395.99 | 699.99 | 129.99 |
| | Mistral | 1382.34 | 692.33 | 126.69 | 1387.98 | 695.23 | 120.35 | 1392.05 | 697.48 | 92.08 | 1388.75 | 695.68 | 108.68 | 1390.98 | 696.63 | 107.37 |
| | Falcon-3 | 1384.25 | 693.37 | 105.03 | 1388.12 | 695.07 | 93.71 | 1389.62 | 695.41 | 93.80 | 1388.77 | 695.67 | 115.24 | 1389.53 | 694.61 | 94.31 |
| | LLaMA-2 | 1384.00 | 692.38 | 91.75 | 1387.93 | 694.93 | 93.30 | 1388.86 | 695.52 | 109.01 | 1388.92 | 694.91 | 100.61 | 1389.93 | 695.33 | 101.76 |
| | LLaMA-3.1 | 1383.84 | 693.10 | 109.09 | 1388.67 | 695.34 | 94.12 | 1389.38 | 695.03 | 93.93 | 1388.72 | 695.19 | 106.55 | 1390.03 | 695.23 | 103.50 |
| | Phi-4 | 1383.99 | 693.18 | 107.90 | 1388.07 | 695.39 | 98.83 | 1389.09 | 695.64 | 108.44 | 1389.50 | 694.70 | 99.72 | 1389.75 | 695.43 | 108.37 |
| | DeepSeek-R1 | 1384.11 | 692.87 | 96.08 | 1388.48 | 695.47 | 102.55 | 1389.54 | 696.04 | 97.25 | 1389.05 | 695.14 | 98.93 | 1389.09 | 694.41 | 95.36 |
| | **STT-LLM** | **1368.99** | **682.16** | **94.92** | **1372.85** | **684.39** | **94.92** | **1374.17** | **684.89** | **94.92** | **1373.51** | **684.05** | **94.91** | **1374.45** | **684.09** | **94.91** |
| Steroid-M$_{lim}$ | Qwen-2.5 | 1750.99 | 901.99 | 106.99 | 1750.99 | 901.99 | 106.99 | 1750.99 | 901.99 | 106.99 | 1750.99 | 901.99 | 106.99 | 1750.99 | 901.99 | 106.99 |
| | Mistral | 1739.59 | 898.42 | 105.45 | 1737.63 | 896.17 | 95.80 | 1742.02 | 899.45 | 103.07 | 1738.92 | 898.42 | 103.42 | 1741.69 | 898.80 | 98.51 |
| | Falcon-3 | 1738.11 | 897.78 | 99.51 | 1738.66 | 898.03 | 102.52 | 1740.75 | 898.42 | 98.74 | 1738.93 | 898.31 | 102.14 | 1742.69 | 900.19 | 97.78 |
| | LLaMA-2 | 1738.57 | 898.38 | 101.46 | 1738.65 | 897.73 | 99.46 | 1741.24 | 898.86 | 100.51 | 1738.90 | 898.48 | 103.02 | 1743.15 | 900.89 | 102.91 |
| | LLaMA-3.1 | 1738.86 | 897.85 | 97.92 | 1737.29 | 896.35 | 100.98 | 1741.25 | 899.54 | 100.01 | 1739.00 | 898.11 | 98.56 | 1743.21 | 900.77 | 98.70 |
| | Phi-4 | 1738.46 | 897.22 | 96.57 | 1738.51 | 898.01 | 102.96 | 1741.42 | 898.43 | 97.71 | 1738.87 | 898.07 | 99.79 | 1743.05 | 900.66 | 98.94 |
| | DeepSeek-R1 | 1737.63 | 897.33 | 98.81 | 1738.12 | 897.42 | 100.40 | 1740.81 | 898.72 | 98.67 | 1739.72 | 898.67 | 99.97 | 1743.62 | 900.59 | 98.15 |
| | **STT-LLM** | **1730.32** | **892.01** | **96.47** | **1730.11** | **891.67** | **96.47** | **1733.18** | **893.01** | **96.47** | **1731.43** | **892.63** | **96.47** | **1734.87** | **894.61** | **96.47** |
| Steroid-F$_{lim}$ | Qwen-2.5 | 1309.99 | 666.99 | 127.99 | 1309.99 | 666.99 | 127.99 | 1309.99 | 666.99 | 127.99 | 1309.99 | 666.99 | 127.99 | 1309.99 | 666.99 | 127.99 |
| | Mistral | 1307.58 | 664.58 | 119.41 | 1292.73 | 657.49 | 123.29 | 1289.67 | 654.38 | 118.12 | 1294.05 | 657.05 | 107.06 | 1286.36 | 652.15 | 126.36 |
| | Falcon-3 | 1305.97 | 662.14 | 96.36 | 1291.65 | 655.82 | 97.87 | 1289.63 | 654.51 | 96.85 | 1294.77 | 656.69 | 102.03 | 1287.89 | 653.04 | 96.47 |
| | LLaMA-2 | 1305.82 | 662.30 | 103.04 | 1292.06 | 656.18 | 100.69 | 1289.05 | 653.63 | 93.96 | 1295.08 | 656.97 | 101.13 | 1287.68 | 654.31 | 106.19 |
| | LLaMA-3.1 | 1306.20 | 663.09 | 106.72 | 1291.13 | 654.67 | 88.66 | 1289.66 | 654.13 | 93.20 | 1293.98 | 656.87 | 115.68 | 1287.32 | 653.76 | 108.22 |
| | Phi-4 | 1306.07 | 662.29 | 99.20 | 1291.92 | 655.84 | 92.49 | 1289.48 | 654.17 | 97.54 | 1294.68 | 656.39 | 102.23 | 1287.37 | 653.85 | 107.41 |
| | DeepSeek-R1 | 1305.70 | 662.66 | 109.88 | 1291.64 | 655.94 | 101.25 | 1289.33 | 654.37 | 103.85 | 1294.89 | 656.59 | 97.21 | 1286.90 | 653.18 | 101.71 |
| | **STT-LLM** | **1290.41** | **651.38** | **94.95** | **1276.32** | **645.16** | **94.92** | **1274.23** | **643.71** | **94.89** | **1279.59** | **645.90** | **94.89** | **1272.19** | **642.90** | **94.86** |

false positives. These results underscore the effectiveness of the structural-temporal tokenization in capturing fine-grained temporal inconsistencies without additional task-specific fine-tuning.

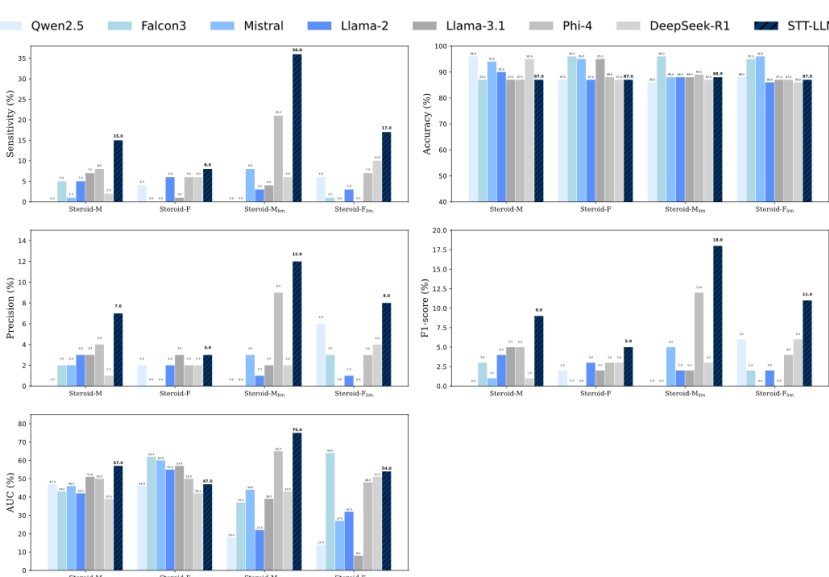

Figure 8: Zero-shot local anomaly detection performance across different metrics.

**Few-Shot Local Anomaly Detection**    Figure 9 shows the few-shot learning performance of all models on local anomaly detection across 2, 5, 10, 15, and 20-shot configurations. STT-LLM demonstrates strong adaptability, achieving the highest or near-highest scores across most metrics and datasets, particularly under 5- and 10-shot settings. Unlike many baselines that fluctuate substantially across shots, STT-LLM maintains stable upward trends in precision, sensitivity, and AUC. For example, in the Steroid-F$_{lim}$ dataset, STT-LLM shows steady improvements in both F1-score and AUC, highlighting its robustness in low-resource regimes. The combination of structural and temporal embeddings appears to improve its generalization in clinical settings with sparse anomaly labels.

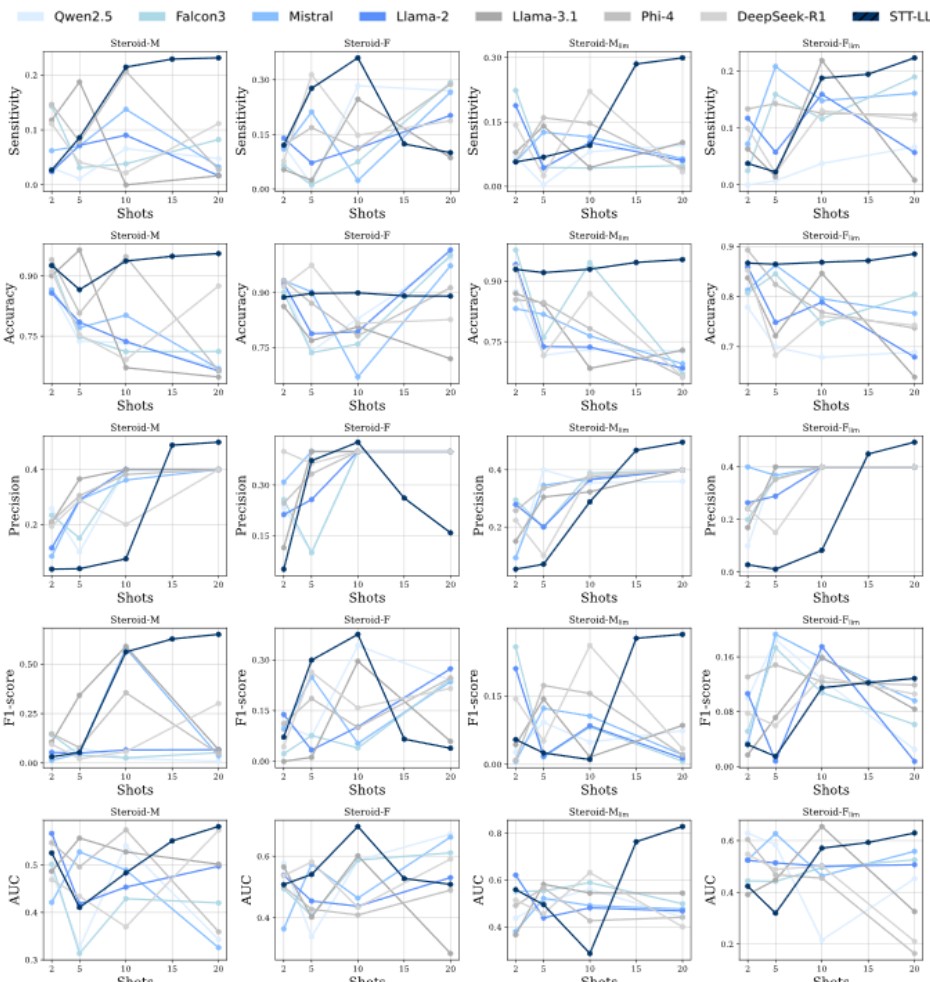

Figure 9: Few-shot local anomaly detection performance across different metrics.

**Zero-Shot Global Anomaly Detection**    In Figure 10, we shows global anomaly detection performance under zero-shot evaluation. STT-LLM outperforms all baselines across most metrics and datasets, achieving the highest F1-scores and AUC on all datasets, including challenging low-data subsets like Steroid-$F_{lim}$. For example, it achieves 26.8% F1-score and 78.6% AUC on Steroid-$F_{lim}$, significantly outperforming the second-best model. Moreover, STT-LLM maintains a strong balance across precision and sensitivity, indicating its ability to detect true anomalies without overfitting to normal patterns. This demonstrates that STT-LLM can generalize effectively even when provided with no additional in-context examples.

**Few-Shot Global Anomaly Detection**    Figure 11 shows the model performance under few-shot global anomaly detection. STT-LLM not only maintains competitive performance in low-shot scenarios but also scales more effectively with additional context compared to baselines. It consistently leads in AUC and F1-score across 10- and 20-shot settings, especially on Steroid-M and Steroid-F datasets. Unlike other models that show unstable or non-monotonic performance trends, STT-LLM improves predictably with more shots, demonstrating strong in-context learning capabilities for rare-event detection. This highlights the strength of its tokenization scheme in enabling efficient information transfer even with minimal labeled supervision.

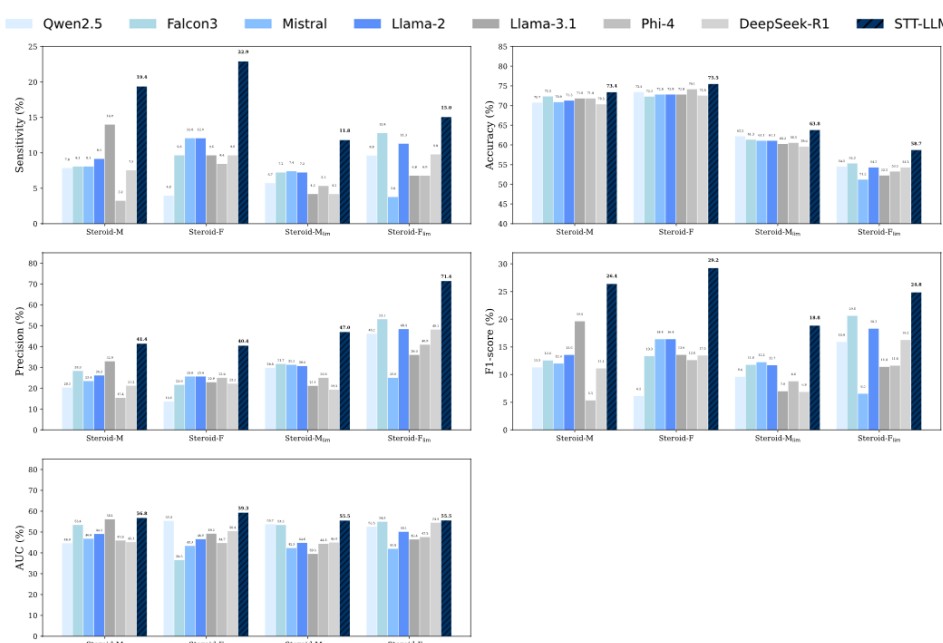

Figure 10: Zero-shot global anomaly detection performance across different metrics.

## A.3 ABLATION STUDIES

To understand the individual contributions of STT-LLM's components, we perform extensive ablation studies across all four datasets. As shown in Tables 8-11, the STT-LLM configuration achieves the best performance across nearly all metrics for both sequence prediction and anomaly detection. When the structural tokenizer is removed, there is a consistent drop in AUC (e.g., from 0.5675 to 0.4964 on Steroid-M, and 0.5927 to 0.5090 on Steroid-F), suggesting that capturing biochemical dependencies between steroid metabolites is important for anomaly discrimination. Similarly, ablating the temporal tokenizer leads to substantial reductions in sensitivity and F1-score, particularly in clinically relevant low-data conditions (e.g., Steroid-$F_{lim}$: F1 drops from 0.2484 to 0.0559).

The projection embedding layer, which aligns structural-temporal features to the LLM-compatible token space, also proves essential. On Steroid-$M_{lim}$, removing embeddings causes the largest AUC drop (0.5548 to 0.5458), and precision degrades across all datasets. Combinations of missing components further compound performance loss. For example, removing both structural and temporal components results in the weakest performance in nearly every metric (e.g., AUC of 0.4877 on Steroid-M and 0.4353 on Steroid-$F_{lim}$). These degradations indicate that no single component is independently sufficient; rather, their integration is key to capturing both temporal variation and physiological structure in a format usable by frozen LLMs. Overall, these ablation results affirm the architectural design of STT-LLM. The synergy between the structural tokenizer (capturing inter-variable relations), temporal tokenizer (capturing temporal evolution), and embedding projection (aligning with LLM input semantics) is important for robust generalization. The consistency of findings across diverse datasets and varying data availability further supports the adaptability and modular design benefits of STT-LLM in real-world clinical anomaly detection and forecasting tasks.

## A.4 CASE STUDY

To evaluate the contextual reasoning ability of STT-LLM, we conducted a controlled comparison against several strong baseline LLMs using a carefully designed prompt. The prompt simulates a real-world scenario where a model should analyze a longitudinal steroid profile consisting of six metabolites measured across multiple time points and answer three questions: (1) identify the anomalous sample, (2) provide a reason based on the steroid metabolism pathway, and (3) determine

Table 8: Contributions of different components in STT-LLM on Steroid-M.

| Model Variants | Sequence Prediction | | | Anomaly Detection (Global) | | | | |
|---|---|---|---|---|---|---|---|---|
| | RMSE↓ | MAE↓ | MAPE↓ | Acc↑ | Sens↑ | Prec↑ | F1↑ | AUC↑ |
| *w/o* all | 1687.71 | 896.39 | 98.93 | 0.7179 | 0.1398 | 0.3291 | 0.1962 | 0.5609 |
| *w/o* structural | 1687.49 | 896.61 | 100.65 | 0.7152 | 0.0968 | 0.2769 | 0.1434 | 0.4964 |
| *w/o* temporal | 1682.45 | 892.85 | 98.38 | 0.7126 | 0.1237 | 0.2987 | 0.1749 | 0.5500 |
| *w/o* embeddings | 1682.75 | 893.40 | 100.56 | 0.7139 | 0.1344 | 0.3125 | 0.1880 | 0.5352 |
| *w/o* structural + temporal | 1682.70 | 893.20 | 98.89 | 0.6967 | 0.0645 | 0.1791 | 0.0949 | 0.4877 |
| *w/o* embeddings + temporal | 1677.56 | 889.29 | 97.07 | 0.7245 | 0.1290 | 0.3429 | 0.1875 | 0.5474 |
| *w/o* embeddings + structural | 1679.16 | 891.78 | 97.35 | 0.7113 | 0.0914 | 0.2576 | 0.1349 | 0.4887 |
| **STT-LLM** | **1664.59** | **881.20** | **96.80** | **0.7338** | **0.1935** | **0.4138** | **0.2637** | **0.5675** |

Table 9: Contributions of different components in STT-LLM on Steroid-F.

| Model Variants | Sequence Prediction | | | Anomaly Detection (Global) | | | | |
|---|---|---|---|---|---|---|---|---|
| | RMSE↓ | MAE↓ | MAPE↓ | Acc↑ | Sens↑ | Prec↑ | F1↑ | AUC↑ |
| *w/o* all | 1384.26 | 693.99 | 114.88 | 0.7280 | 0.0964 | 0.2286 | 0.1356 | 0.4919 |
| *w/o* structural | 1368.79 | 682.52 | 97.64 | 0.7467 | 0.1687 | 0.3500 | 0.2276 | 0.5090 |
| *w/o* temporal | 1369.23 | 683.17 | 101.36 | 0.7280 | 0.0620 | 0.1724 | 0.0893 | 0.4682 |
| *w/o* embeddings | 1383.83 | 693.49 | 104.61 | 0.7413 | 0.1807 | 0.3409 | 0.2362 | 0.4998 |
| *w/o* structural + temporal | 1373.63 | 686.26 | 99.16 | 0.7307 | 0.0482 | 0.1538 | 0.0734 | 0.4017 |
| *w/o* embeddings + temporal | 1380.30 | 690.67 | 95.46 | 0.7493 | 0.1928 | 0.3721 | 0.2540 | 0.5909 |
| *w/o* embeddings + structural | 1384.13 | 693.57 | 100.04 | 0.7333 | 0.1205 | 0.2703 | 0.1667 | 0.4663 |
| **STT-LLM** | **1368.44** | **682.39** | **94.92** | **0.7547** | **0.2289** | **0.4043** | **0.2923** | **0.5927** |

Table 10: Contributions of different components in STT-LLM on Steroid-M$_{\text{lim}}$.

| Model Variants | Sequence Prediction | | | Anomaly Detection (Global) | | | | |
|---|---|---|---|---|---|---|---|---|
| | RMSE↓ | MAE↓ | MAPE↓ | Acc↑ | Sens↑ | Prec↑ | F1↑ | AUC↑ |
| *w/o* all | 1737.36 | 897.95 | 102.07 | 0.6024 | 0.0418 | 0.2115 | 0.0698 | 0.3949 |
| *w/o* structural | 1737.92 | 897.88 | 100.73 | 0.6214 | 0.0570 | 0.3261 | 0.0971 | 0.5109 |
| *w/o* temporal | 1732.97 | 894.13 | 98.19 | 0.6174 | 0.0913 | 0.3582 | 0.1455 | 0.4199 |
| *w/o* embeddings | 1733.26 | 894.47 | 98.78 | 0.6269 | 0.0993 | 0.4250 | 0.1783 | 0.5458 |
| *w/o* structural + temporal | 1733.22 | 893.20 | 98.89 | 0.6159 | 0.0645 | 0.1791 | 0.0949 | 0.4877 |
| *w/o* embeddings + temporal | 1732.17 | 892.56 | 96.78 | 0.6364 | 0.0038 | 0.1429 | 0.0074 | 0.3759 |
| *w/o* embeddings + structural | 1731.42 | 892.27 | 97.09 | 0.6119 | 0.0608 | 0.2909 | 0.1006 | 0.4373 |
| **STT-LLM** | **1730.04** | **892.08** | **96.46** | **0.6377** | **0.1179** | **0.4697** | **0.1884** | **0.5548** |

Table 11: Contributions of different components in STT-LLM on Steroid-F$_{\text{lim}}$.

| Model Variants | Sequence Prediction | | | Anomaly Detection (Global) | | | | |
|---|---|---|---|---|---|---|---|---|
| | RMSE↓ | MAE↓ | MAPE↓ | Acc↑ | Sens↑ | Prec↑ | F1↑ | AUC↑ |
| *w/o* all | 1305.98 | 662.99 | 100.47 | 0.5222 | 0.0677 | 0.3600 | 0.1139 | 0.4644 |
| *w/o* structural | 1301.38 | 651.91 | 97.40 | 0.5085 | 0.0376 | 0.2381 | 0.0649 | 0.4056 |
| *w/o* temporal | 1301.44 | 652.57 | 101.23 | 0.5392 | 0.0301 | 0.4000 | 0.0559 | 0.4946 |
| *w/o* embeddings | 1305.87 | 662.88 | 104.29 | 0.5392 | 0.0977 | 0.4643 | 0.1615 | 0.5090 |
| *w/o* structural + temporal | 1301.52 | 655.66 | 99.21 | 0.5222 | 0.0226 | 0.2308 | 0.0411 | 0.4353 |
| *w/o* embeddings + temporal | 1302.30 | 660.06 | 95.61 | 0.5324 | 0.0902 | 0.4286 | 0.1491 | 0.4128 |
| *w/o* embeddings + structural | 1306.18 | 662.97 | 100.18 | 0.5392 | 0.0301 | 0.4000 | 0.0559 | 0.4792 |
| **STT-LLM** | **1301.24** | **651.79** | **94.97** | **0.5870** | **0.1504** | **0.7143** | **0.2484** | **0.5555** |

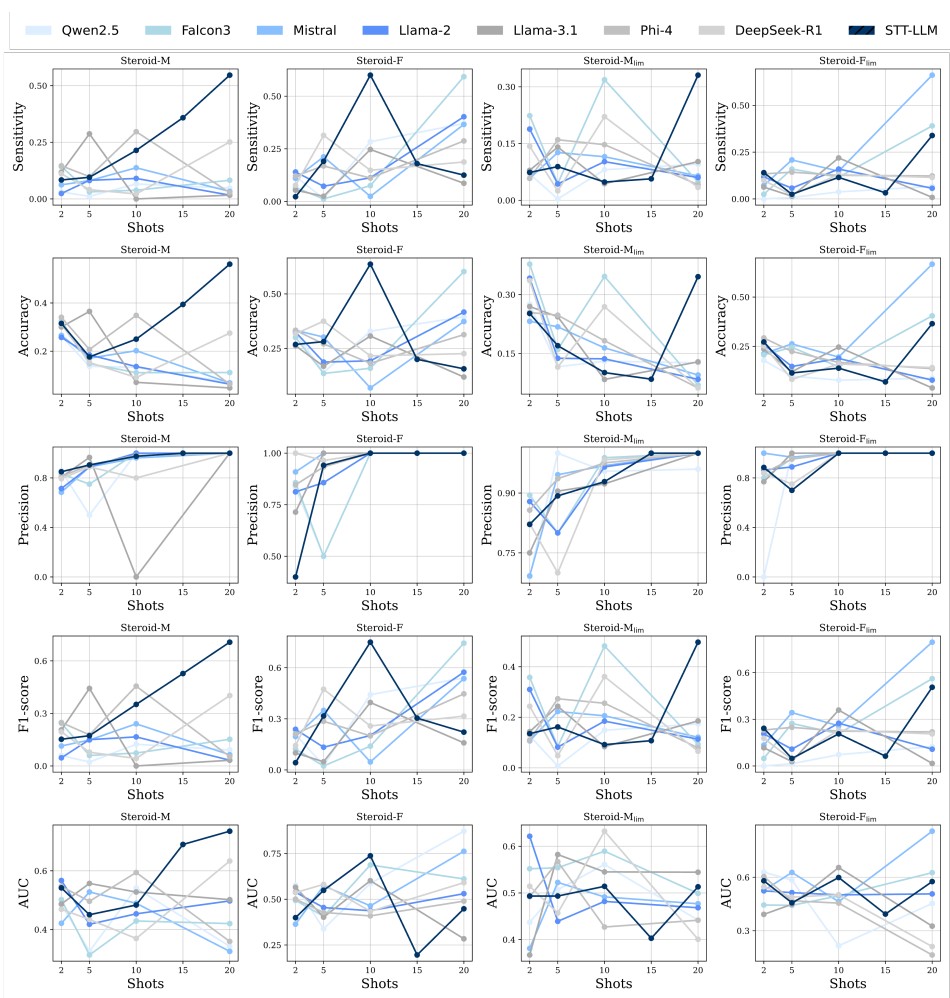

Figure 11: Few-shot global anomaly detection performance across different metrics.

whether the anomaly is due to steroid doping or a mismatched sample. Figure 12 shows the generated responses of different LLMs on a profile with a known anomaly (Sample 4) caused by steroid doping.

As shown in the responses, most LLMs either failed to detect the correct anomaly or provided vague and incoherent explanations. For example, Phi and Qwen misattributed the anomaly to sample swapping despite correctly identifying the anomalous sample. DeepSeek failed to reach a decisive conclusion, and Falcon made inconsistent inferences based on isolated metabolite values. Although Mistral and LLaMA-3 correctly identified Sample 4, their reasoning relied on static domain knowledge (e.g., typical metabolite levels) rather than relational comparison across samples in the profile. This limitation arises because general-purpose LLMs are not inherently equipped to model temporal dynamics or biochemical dependencies across multivariate time-series.

In contrast, STT-LLM correctly identified the anomaly and generated a structured explanation grounded in structural-temporal relationships (Figure 13). It reasoned that the value of metabolite T (29.0) in Sample 4 was significantly higher than in the other samples (range 12-20). It further justified that normal metabolic progression could not explain this deviation. Instead, it correctly attributed the anomaly to potential doping. This capability highlights the value of STT-LLM's embedding-guided tokenization in aligning metabolite patterns over time and across variables, something missing in models lacking structured input alignment. Importantly, this structural-temporal reasoning also allows STT-LLM to avoid false positives in edge cases, such as when all samples from a subject appear uniformly elevated due to a legitimate medical condition. While other LLMs might misclassify such

cases as doping, STT-LLM has the capacity to model intra-profile consistency, thereby reducing bias and increasing interpretability in high-stakes clinical scenarios.

**Phi**

1) The anomalous sample is Sample 9.

2) The reason is that Sample 9 shows a significantly higher concentration of metabolites Adiol, Bdiol, A, Etio, and T, which deviates from the expected pattern of steroid metabolism.

3) The anomaly is likely because the sample does not belong to the same athlete, as the metabolite concentrations are inconsistent with the rest of the profile.

6) The anomaly suggests the presence of a sample from a different source rather than a pattern of steroid use.

**Llama3**

1. The anomalous sample is the one where Adiol is much higher than the other steroid metabolites, which is sample 4.

2. The reason for the anomaly in sample 4 is likely due to the high levels of Etio and Adiol, indicating that the athlete may have been taking anabolic steroids or undergoing intense steroid use, causing an imbalance in the steroid metabolism pathway.

3. The anomaly is likely due to steroid doping because the high levels of Etio and Adiol are not typical of a natural steroid profile.

**DeepSeek**

Maybe the issue is that the diols are lower than expected? Or that T is low relative to A, which would be a problem because normally, T should be higher than A if there's metabolism. Or wait, A is the precursor to T, so A should be lower when T is high. If A is high and T is low, that would be unusual. So in samples 1 and 2, A is high, T is low. The same for sample 3: A is 820, T is 12, so A/T is about 68:1. Again, A is much higher than T.

**Mistral**

1) Sample 4

2) Anomaly in Sample 4: In sample 4 the concentration of Bdiol is 160 ug/ml which is not consistent with the expected concentration of Bdiol based on the steroid metabolism pathway. The concentration of Bdiol should be 1.6ug/ml (160/100) and hence the sample is anomalous.

**Falcon**

1) Sample 9: The metabolite A is 3000.0 and the metabolite Etio is 2100.0. The metabolite A should be higher than the metabolite Etio as it comes after the Bdiol in the pathway

2) The sample number of the anomalous sample is 9

3) I think this sample is due to steroid doping

**Qwen**

Q: 1)Sample 9 is the anomalous sample within the profile.

2) A: Sample 9 shows an unusually high value for both Adiol and Bdiol, indicating a possible metabolic anomaly or potential steroid doping.

3) The anomaly in sample 9 is most likely due to steroid doping, as the unusually high values for Adiol and Bdiol suggest an external source of these metabolites.

Figure 12: Contextual explanations from different models.

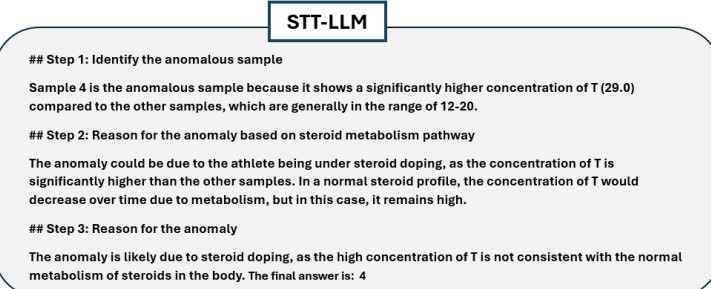

**STT-LLM**

## Step 1: Identify the anomalous sample

Sample 4 is the anomalous sample because it shows a significantly higher concentration of T (29.0) compared to the other samples, which are generally in the range of 12-20.

## Step 2: Reason for the anomaly based on steroid metabolism pathway

The anomaly could be due to the athlete being under steroid doping, as the concentration of T is significantly higher than the other samples. In a normal steroid profile, the concentration of T would decrease over time due to metabolism, but in this case, it remains high.

## Step 3: Reason for the anomaly

The anomaly is likely due to steroid doping, as the high concentration of T is not consistent with the normal metabolism of steroids in the body. The final answer is: 4

Figure 13: Contextual explanations from STT-LLM model.

