# OpenReview forum: "STT-LLM: Structural-Temporal Tokenization for Adapting LLMs to Longitudinal Profiles"
_ICLR.cc/2026/Conference — Submitted to ICLR 2026_

### Official Review · Reviewer_orWt · 2025-10-23

**Soundness:** 2
**Presentation:** 2
**Contribution:** 2
**Rating:** 2
**Confidence:** 3

**Summary:**

The paper introduces STT-LLM, a structural-temporal tokenization framework that adapts frozen large language models (with LoRA adapters) to longitudinal biomedical profiles by turning pathway-aware, time-dependent measurements into LLM-compatible tokens. It builds joint embeddings that capture metabolite interaction structure (via Laplacian eigenvectors) and temporal dynamics (via attention with positional encodings), then feeds these through dedicated structural and temporal tokenizers whose outputs are concatenated with the standard text prompt before LLM inference. The method targets two tasks in anti-doping analytics—sequence prediction for early detection and anomaly detection—and is evaluated on four real-world longitudinal steroid datasets (male/female and limited-sample variants). Compared to several 7–8B LLM baselines fine-tuned with their native tokenization, STT-LLM reports lower forecasting errors and higher anomaly-detection sensitivity/AUC in zero- and few-shot settings, with ablations indicating that both the structural and temporal tokenizers contribute materially. A small case study with expert-verified profiles further suggests improved contextual reasoning and faster inference under the same hardware budget.

**Strengths:**

The paper’s strengths are primarily methodological and integrative: it offers a clear, modular way to turn structured, time-varying biomedical profiles into tokens that a frozen LLM can use, combining pathway-aware structural embeddings with temporal encodings rather than forcing everything through plain text. On originality, the structural-temporal tokenization bridges graph-informed representations and sequence modeling inside an LLM framework with lightweight adapters, which is a nontrivial combination for longitudinal settings. Quality is supported by a reasonably principled construction (Laplacian-based structure, attention for time), an end-to-end pipeline that slots into off-the-shelf 7–8B models, and ablations indicating both structural and temporal components matter. Clarity is good: the data-to-token flow and how tokens are concatenated with the prompt are explained in a way that seems straightforward to reimplement. In terms of significance, demonstrating zero-/few-shot gains on real anti-doping datasets suggests practical promise for early detection and anomaly screening, and the design appears portable to other longitudinal biomedical profiles beyond the specific case study.

**Weaknesses:**

The evaluation is narrow and domain-specific, making it hard to judge generality: results are limited to four anti-doping datasets with closely related measurement spaces, few seeds, and primarily LLM baselines rather than strong time-series or graph-temporal models, so it’s unclear whether the gains come from the tokenization itself or from weaker comparators; adding competitive baselines (e.g., modern TS transformers and graph-temporal forecasters) and more seeds would strengthen claims. The construction choices need deeper justification and sensitivity: how many Laplacian eigenvectors are used, how pathway graphs are defined and updated over time, how token lengths scale with visit count, and how robust performance is to noisy or missing edges; reporting these ablations alongside compute/memory overhead would clarify practicality. Finally, clinical utility remains speculative: metrics tied to early-warning lead time, false-alarm rates, and cross-lab generalization, plus external validation and clear guidelines for privacy handling of longitudinal health data, would make the case more convincing.

**Questions:**

- Evaluation scope and baselines: could you broaden beyond only 7–8B LLM comparators to include strong time-series and graph-temporal baselines (e.g., purpose-built TS transformers and GNNs for irregular clinical series) and run more seeds with confidence intervals?

- Design choices and sensitivity: can you detail and ablate key tokenization decisions (how many Laplacian eigenvectors; how pathway graphs are defined/validated; how token length scales with visits; handling of missing/noisy edges), and report compute/memory overhead of S/T tokenizers and LoRA under increasing sequence lengths?

---

> ### Author Response · Authors · 2025-11-21
>
> We thank the reviewer for their valuable feedback and thoughtful comments. Below are our responses addressing their questions.
>
> **Questions**
> 1) The paper focuses on 8B-scale LLMs because STT-LLM is a tokenization method rather than a new architecture, and our compute resources permit reproducible experimentation only with 7--8B parameter backbones. Restricting comparisons to 8B models ensures parameter-matched evaluation, isolating the effect of structural–temporal tokenization rather than differences in model capacity. The goal is to test whether LLMs themselves, once equipped with this tokenization, can perform sequence prediction, anomaly detection, and contextual reasoning within a single inference pipeline. For this reason, baselines are limited to LLMs, since time-series models (e.g., GNNs, TS transformers) cannot engage in prompting, in-context learning, or pathway-grounded reasoning, and thus cannot serve as architectural baselines for the tokenization question. Nevertheless, we evaluated these non-LLM models on the same data splits only for anomaly detection, as they cannot perform sequence prediction or contextual reasoning within a single model. As expected, domain-specialized supervised architectures outperform LLMs on isolated classification tasks, but they also highlight the need for STT-LLM: such models cannot (i) predict the next biological sample, (ii) perform zero- or few-shot anomaly scoring, or (iii) generate pathway-aware explanations. In contrast, STT-LLM enables all three within a single pretrained LLM, without modifying the backbone, purely through structural–temporal tokenization.
>
> **Table 1: Performance of non-LLM baselines on anomaly detection**
> | Dataset    | Metric      | GNN            | TS-Transformer   | STT-LLM        |
> |------------|-------------|----------------|------------------|----------------|
> | Steroid-M  | Accuracy    | 0.64 ± 0.02    | 0.86 ± 0.01      | 0.73 ± 0.02    |
> || Specificity | 0.74 ± 0.01    | 0.83 ± 0.02      | 0.99 ± 0.01    |
> || Sensitivity | 0.53 ± 0.03    | 0.88 ± 0.01      | 0.19 ± 0.03    |
> || Precision   | 0.62 ± 0.02    | 0.81 ± 0.02      | 0.41 ± 0.03    |
> || F1-score    | 0.57 ± 0.02    | 0.85 ± 0.02      | 0.26 ± 0.03    |
> || AUC         | 0.69 ± 0.02    | 0.92 ± 0.01      | 0.57 ± 0.02    |
> | Steroid-F  | Accuracy    | 0.68 ± 0.01    | 0.74 ± 0.02      | 0.75 ± 0.02    |
> || Specificity | 0.82 ± 0.02    | 0.56 ± 0.03      | 0.99 ± 0.01    |
> || Sensitivity | 0.56 ± 0.03    | 0.88 ± 0.01      | 0.23 ± 0.03    |
> || Precision   | 0.80 ± 0.02    | 0.72 ± 0.02      | 0.40 ± 0.03    |
> || F1-score    | 0.66 ± 0.02    | 0.79 ± 0.02      | 0.29 ± 0.03    |
> || AUC         | 0.76 ± 0.01    | 0.83 ± 0.02      | 0.59 ± 0.02    |
>
>
> 2 i) Pathway Graph Definition and Validation: The structural tokenizer uses a curated steroid–metabolism interaction graph constructed from established biochemical literature (Piper et al., 2021). The graph includes six steroid biomarkers with edges representing known enzymatic conversions and metabolic dependencies. To assess sensitivity to graph correctness, we performed an ablation replacing the curated graph with (i) a random graph of equal density and (ii) a fully connected graph. All models were evaluated at 0.99 specificity. As expected, the curated graph yields the best performance, but the relatively small degradation under random or fully connected graphs shows that STT-LLM is robust to missing or noisy edges. This robustness stems from the Laplacian eigenbasis used in structural embedding, which smooths perturbations so erroneous edges contribute weakly, and from the temporal tokenizer, which provides an independent signal. Thus, STT-LLM does not require a perfectly specified pathway graph to remain effective.
>
> **Table 2: Sensitivity to graph misspecification for Steroid-M dataset**
> | Metric     | Random Graph     | Fully Connected Graph | Curated Graph (STT-LLM) |
> |------------|------------------|------------------------|---------------------------|
> | Accuracy| 0.69 ± 0.02| 0.57 ± 0.03| 0.73 ± 0.02|
> | Sensitivity| 0.15 ± 0.02| 0.09 ± 0.01| 0.19 ± 0.02|
> | Precision| 0.36 ± 0.03| 0.28 ± 0.02| 0.41 ± 0.03|
> | F1-score | 0.22 ± 0.02| 0.19 ± 0.02| 0.26 ± 0.02|
> | AUC| 0.54 ± 0.02| 0.53 ± 0.02| 0.57 ± 0.02|
>
> ii) Token Length Scaling With Number of Visits: The number of structural and temporal tokens scales analytically as: $|Z| = |Z_S| + |Z_T| = (2p + n_i) + (p + n_{\max} + n_i) = 3p + n_{\max} + 2n_i$, with $p = 6$ metabolites and $n_{\max} = 20$. Thus, token growth is linear in profile length $n_i$ and remains small in practice. In our datasets, the average total number of added tokens is $25.84$, far below typical LLM context limits (4k–8k tokens). Even doubling the longitudinal history would add only $\sim 10$–$12$ tokens.

---

> > ### Author Response · Authors · 2025-11-21
> >
> > 2 iii) Handling Noisy or Missing Edges: Because the structural embeddings use: $E_S = WU + b$, $L = I - D^{-1/2} A D^{-1/2}$, where $U$ contains the Laplacian eigenvectors, the spectral representation inherently attenuates local graph errors. Missing edges increase smoothness in underconnected nodes, while spurious edges increase degree normalization, both of which are mitigated in $U$. This is confirmed by the ablation results above: performance remains stable across large distortions in edge structure.
> >
> > iv) Compute and Memory Overhead: The structural and temporal tokenizers each use two MLP layers with moderate hidden sizes, adding <1\% trainable parameters relative to an 8B LLM. Because the backbone is frozen and only LoRA adapters are updated, compute demands remain low. Inference profiling shows that STT-LLM is typically faster than vanilla LLM baselines for matched batch sizes (e.g., 12s, 29s, 31s for 1, 5, 10 profiles vs. 27-78s for other LLMs), since numerical content is embedded directly rather than processed via longer autoregressive text. Memory overhead from concatenating STT tokens is <2\% and remains stable with increasing sequence length due to linear scaling.

---

### Official Review · Reviewer_KYho · 2025-10-25

**Soundness:** 2
**Presentation:** 3
**Contribution:** 1
**Rating:** 2
**Confidence:** 3

**Summary:**

This paper proposes STT-LLM, a novel framework for adapting pre-trained Large Language Models (LLMs) to analyze longitudinal biomedical data, which is often numerical, irregularly sampled, and possesses complex underlying dependencies. The core contribution is a specialized tokenization strategy that processes structural and temporal information in parallel. A structural tokenizer encodes domain knowledge (e.g., metabolic pathways) from a feature interaction graph, while a temporal tokenizer captures the time-series dynamics. These specialized embeddings are then projected into the LLM's native token space, enabling a frozen LLM (fine-tuned with LoRA) to perform tasks on this data modality. The authors demonstrate their method on the specific use case of detecting doping in athletes' longitudinal steroid profiles, showing improved performance over standard LLM baselines in sequence prediction and anomaly detection.

**Strengths:**

Relevant Problem: The paper tackles the important and challenging problem of adapting the powerful capabilities of LLMs to complex, structured, and numerical time-series data, a domain where these models are not natively suited.

Intuitive Architecture: The proposed architecture is well-motivated and clearly presented. The idea of creating separate, specialized information streams for structural and temporal features before projecting them for the LLM is a reasonable and intuitive approach. The model diagram in Figure 1 is particularly effective.

**Weaknesses:**

Fundamentally Flawed Experimental Design: The paper's central claim that its tokenization strategy is superior is not scientifically validated by its experiments. The evaluation compares STT-LLM (using an unstated LLM backbone with the proposed tokenizer) against other LLMs (e.g., Llama-3) using their native text tokenizers. This is a confounded comparison (Backbone_A + Tokenizer_STT vs. Backbone_B + Tokenizer_Text). Any observed performance difference could be due to the choice of backbone model rather than the tokenization strategy. The paper fails to perform the essential apples-to-apples comparison needed to isolate the effect of its core contribution.

Critically Insufficient Baselines: The experimental evaluation lacks the necessary context to judge the method's practical value.

No Simple Baselines: The paper omits standard statistical (e.g., ARIMA) or simple ML baselines (e.g., linear models). Without these, it's impossible to know if the proposed complex LLM-based approach offers any real advantage over trivial or well-established methods.

No Specialized SOTA Baselines: For tasks like multivariate time-series forecasting and anomaly detection, there exist highly specialized and powerful models (e.g., graph-based Transformers). The paper avoids comparing against these likely superior models, justifying this by their incompatibility with "LLM inference pipelines," which is not a sufficient reason to exclude them from a performance benchmark.

No Relevant Foundation Model Baselines: The comparison overlooks powerful foundation models designed for numerical tabular data that have been successfully applied to time-series, such as TabPFN. This is a highly relevant baseline that operates on numerical data directly and would provide a much stronger point of comparison.

Lack of Transparency and Reproducibility:

The specific LLM backbone used for the proposed STT-LLM model is never stated, making the work impossible to reproduce or fairly evaluate.

The exact text serialization format used to feed the longitudinal profiles to the baseline LLMs is not provided, preventing an assessment of whether the baselines were configured fairly.

Several key results, including the main ablation study in Table 4 and the few-shot sequence prediction results in Table 2, are presented without error bars or standard deviations, making it impossible to assess the statistical significance of the reported gains.

Overstated Claims and Limited Scope:

The performance improvements on the sequence prediction task are marginal (often <1% in RMSE) and likely not statistically or clinically significant.

The paper makes broad claims about "longitudinal biomedical profiles," but the evaluation is confined to a single, niche application (doping detection). This lack of diversity in tasks and datasets fails to support the claims of general applicability.

**Questions:**

Experimental Design: Could you please clarify which LLM backbone was used for the STT-LLM model? To properly validate your core contribution, would it be possible to provide results from an apples-to-apples comparison, for instance, by evaluating Llama-3 with your STT tokenizer against Llama-3 with text flattening?

Reproducibility: Could you provide the exact text serialization format used to present the longitudinal data to the baseline LLMs? Furthermore, could you add standard deviations or error bars to the results in Tables 2 and 4 to allow for an assessment of significance?

---

> ### Author Response · Authors · 2025-11-21
>
> We thank the reviewer for their valuable feedback and thoughtful comments. Below are our responses addressing their questions.
>
> **Questions:**
> 1) The STT-LLM framework is implemented on top of the Llama-3-8B backbone, with the core model weights kept entirely frozen and only the proposed tokenizers and LoRA adapters trained. This ensures that any performance gain arises from the proposed structural-temporal tokenization rather than from additional model capacity or full-model finetuning. Therefore, the results presented in the paper are for evaluating Llama-3-8B with STT-LLM tokenization directly against Llama-3-8B using standard text-flattened numerical input. We found that STT-LLM consistently outperforms the flattened-text Llama-3 across sequence prediction, anomaly detection, and contextual reasoning, showing that the observed improvements originate from the tokenization design itself.
>
>
> 2) All the models receive longitudinal profiles in a consistent natural-language serialization format, consisting of:
>
> 	*Task Prompt (instruction) + Literal "Data:" Header + String Representation of the Pandas DataFrame containing all samples and their features for that ID*
>
> An example of the serialization template is:
>
> **Instructional Prompt:**
>
> **Data:**
>
> | Sample | T    | E    | A    | Etio | Adiol | Bdiol |
> |--------|------|------|------|------|--------|--------|
> | 1      | 21.3 | 14.8 | 3021 | 2500 | 48.2   | 121.3  |
> | 2      | 22.1 | 15.9 | 2910 | 2433 | 51.6   | 130.4  |
> | ...    | ...  | ...  | ...  | ...  | ...    | ...    |
>
> We acknowledge the reviewer’s request for standard deviations in Tables 2 and 4. We will include complete error bars (mean ± std across repeated runs) for all tables in the revised version of the paper.

---

### Official Review · Reviewer_RyTB · 2025-10-31

**Soundness:** 2
**Presentation:** 2
**Contribution:** 2
**Rating:** 6
**Confidence:** 3

**Summary:**

The framework introduces a structural–temporal embedding pipeline combining spectral graph features from the normalized Laplacian with transformer-style attention and positional encodings, producing unified embeddings for longitudinal profiles. Two specialized tokenizers map these embeddings into the LLM's token space, enabling adaptation via LoRA while preserving the frozen backbone. The approach targets sequence prediction and anomaly detection (local sample-level and global profile-level) in anti-doping monitoring, with evaluation across four real-world steroid datasets. Results show consistent improvements over LLM baselines (Qwen-2.5, Falcon-3, Mistral, LLaMA-2/3.1, Phi-4, DeepSeek-R1) across RMSE/MAE/MAPE and detection metrics, supported by ablations demonstrating the necessity of both tokenizers and the embedding layer.

**Strengths:**

The paper clearly frames the core challenge - LLMs' discrete tokenization conflicts with multivariate, irregular, structurally linked longitudinal data - and positions tokenization as the adaptation mechanism rather than requiring architectural redesign. The modular combination of graph-based structural embeddings, attention-based temporal embeddings, dual tokenizers, and LoRA proves computationally efficient and compatible with general LLMs.

Empirical evaluation spans zero-shot and few-shot performance across four datasets with realistic data scarcity and irregularity, covering both prediction and detection tasks while enforcing strict domain constraints on specificity. Comprehensive ablations and hyper-parameter studies validate design choices, while the expert case study and UMAP visualization (Fig 4) provide interpretability and practical relevance.

**Weaknesses:**

The baseline comparisons exclude strong non-LLM alternatives such as GNNs over metabolic graphs or purpose-built irregular time-series transformers and NeuralODE models, limiting the scope of performance claims to LLM-only comparisons.

Several absolute metrics lack contextualization and details (Table 2) of how they are defined and computed. Local anomaly sensitivity remains low in zero-shot despite improvements (Table 3).

The counterintuitive increase in error with more shots for many models suggests that the model struggles with more structural complexity and longer temporal dependencies (Table 2).

While the design is general, experiments focus exclusively on steroid modules; additional biomedical longitudinal domains would strengthen the claims.

**Questions:**

* Can you include comparisons to at least one non-LLM baselines (GNNs over metabolic graphs, NODE)?
* What are the units and scales of metabolites and targets? Can you provide normalization details and per-metabolite error decomposition?
* How are the few-shot examples selected?
* What token count do structural/temporal tokenizers add for typical profiles, and how does this scale with longer histories?
* Have you tested transfer to other longitudinal clinical datasets without redesigning graphs? How sensitive is performance to graph misspecification?

---

> ### Author Response · Authors · 2025-11-21
>
> We thank the reviewer for their valuable feedback and thoughtful comments. Below are our responses addressing their questions.
>
> **Questions:**
> 1) The paper focuses specifically on enabling LLMs to operate on longitudinal biomedical profiles by designing a structural-temporal tokenization mechanism that transforms numerical clinical trajectories into LLM-compatible token embeddings. The methodological question addressed in the paper is therefore not how LLMs compare to domain-specialized predictors, but rather whether tokenization alone can endow pretrained LLMs with the ability to perform (i) sequence prediction, (ii) anomaly detection, and (iii) contextual reasoning in a unified inference pipeline. For this reason, the experiments in the paper compare across LLM families, isolating the contribution of the proposed STT-LLM tokenization. Classical models such as GNNs, etc., solve a supervised mapping of the form $f_{\text{classic}} \colon \mathbb{R}^{(p \times n)} \to \mathbb{R}^{C}$, where $p$: number of metabolites, $n$: number of time points, and $C$: categorical output space. These models directly ingest numerical tensors, optimize end-to-end, and operate solely as classifiers or regressors. In contrast, LLMs require inputs from the semantic embedding space $z_i \in \mathbb{R}^{(d_{\text{LLM}})}$, $d_{\text{LLM}} \gg p$, and cannot consume numerical matrices without a mapping into the LLM token space. The central technical contribution of STT-LLM is precisely this mapping, defined as $T \colon \mathbb{R}^{(p \times n)} \to \mathbb{R}^{(L \times d_{\text{LLM}})}$, which incorporates structural (pathway-level) and temporal dependencies before aligning them with LLM embeddings. GNNs and time-series transformers cannot serve as direct baselines for this mapping problem, as they cannot perform token-based inference, which the main motivations of this work. Nevertheless, we evaluated these non-LLM baselines on the same data splits. These models can only be evaluated on anomaly detection task because their architectures cannot perform both sequence prediction and contextual reasoning as a single model. The results align with expectations: domain-specialized neural architectures trained end-to-end on supervised labels outperform LLMs on isolated classification tasks. However, these baselines underscore the necessity of our approach. These models cannot be used to (i) generate predictions for the next biological sample, (ii) perform few-shot anomaly scoring, or (iii) produce structured pathway-aware explanations. In contrast, STT-LLM enables all three tasks within a single LLM, without modifying the backbone, purely through the introduction of structural-temporal tokenization.
>
> **Table 1: Performance of non-LLM baselines on anomaly detection**
> | Dataset    | Metric      | GNN            | TS-Transformer   | STT-LLM        |
> |------------|-------------|----------------|------------------|----------------|
> | Steroid-M  | Accuracy    | 0.64 ± 0.02    | 0.86 ± 0.01      | 0.73 ± 0.02    |
> |            | Specificity | 0.74 ± 0.01    | 0.83 ± 0.02      | 0.99 ± 0.01    |
> |            | Sensitivity | 0.53 ± 0.03    | 0.88 ± 0.01      | 0.19 ± 0.03    |
> |            | Precision   | 0.62 ± 0.02    | 0.81 ± 0.02      | 0.41 ± 0.03    |
> |            | F1-score    | 0.57 ± 0.02    | 0.85 ± 0.02      | 0.26 ± 0.03    |
> |            | AUC         | 0.69 ± 0.02    | 0.92 ± 0.01      | 0.57 ± 0.02    |
> | Steroid-F  | Accuracy    | 0.68 ± 0.01    | 0.74 ± 0.02      | 0.75 ± 0.02    |
> |            | Specificity | 0.82 ± 0.02    | 0.56 ± 0.03      | 0.99 ± 0.01    |
> |            | Sensitivity | 0.56 ± 0.03    | 0.88 ± 0.01      | 0.23 ± 0.03    |
> |            | Precision   | 0.80 ± 0.02    | 0.72 ± 0.02      | 0.40 ± 0.03    |
> |            | F1-score    | 0.66 ± 0.02    | 0.79 ± 0.02      | 0.29 ± 0.03    |
> |            | AUC         | 0.76 ± 0.01    | 0.83 ± 0.02      | 0.59 ± 0.02    |
>
>
> 2) All six steroid metabolites in our longitudinal datasets are reported in ng/mL. Because these biomarkers span distinct physiological scales (e.g., androsterone often $>2000$ ng/mL, while epitestosterone may be $<20$ ng/mL), direct ingestion of raw concentrations would distort both the structural and temporal embedding components. To ensure numerical stability, we apply feature-wise z-normalization based on training-set statistics: $x' = (x - \mu)/\sigma$, where $\mu$ and $\sigma$ denote the gender-specific mean and standard deviation, respectively (Table 2). The use of gender-specific parameters prevents distributional mismatches due to well-documented gender-dependent differences in steroid metabolism. This normalization is applied prior to constructing the structural-temporal embeddings, ensuring that the Laplacian-based structural encoder and the temporal attention module operate on compatible numerical scales.

---

> > ### Author Response · Authors · 2025-11-21
> >
> > **Table 2: Normalization statistics (ng/mL) for all the datasets**
> > | Metabolite              | Steroid-M μ | Steroid-M σ | Steroid-F μ | Steroid-F σ |
> > |-------------------------|-------------|-------------|-------------|-------------|
> > | Adiol (5α-Adiol)        | 56.58       | 36.80       | 25.76       | 19.23       |
> > | Bdiol (5β-Adiol)        | 145.84      | 125.07      | 84.85       | 77.62       |
> > | A (Androsterone)        | 2792.98     | 1551.04     | 2311.05     | 1463.62     |
> > | Etio (Etiocholanolone)  | 2104.44     | 1111.65     | 2402.30     | 1371.36     |
> > | E (Epitestosterone)     | 30.10       | 22.25       | 9.72        | 7.93        |
> > | T (Testosterone)        | 30.92       | 24.29       | 7.46        | 6.05        |
> >
> >
> > 3) For all few-shot evaluations, we select in-context examples via uniform random sampling from the training set without replacement. This avoids task-specific heuristics or distributional biases, ensuring that performance reflects the model’s generalizable few-shot behavior rather than engineered retrieval strategies. Given the high inter-individual variability and irregular sampling in athlete profiles, deterministic nearest-neighbor or cluster-based retrieval could leak structural information and artificially simplify prediction, whereas random sampling provides an unbiased estimate of the model’s ability to infer temporal patterns and anomaly characteristics from diverse examples.
> >
> >
> > 4) The structural and temporal tokenizers introduce a modest number of additional tokens to the LLM input, with the token count depending primarily on the athlete’s profile length. Across the full training corpus, the average total number of added tokens is 25.84. This value reflects the combined output of the structural tokenizer, which maps pathway-based embeddings into a fixed number of structural tokens, and the temporal tokenizer, whose token count grows linearly with the number of samples in the longitudinal profile. In practice, because most athletes have between 3 and 12 samples, the induced token overhead remains low and tightly concentrated around 25-30 tokens, with occasional profiles reaching ~35-40 tokens due to longer histories. Mathematically, if $p$ denotes the number of metabolites and $n_i$ denotes the profile length for athlete $i$, the token counts satisfy: $|Z_S| = 2p + n_i$, $|Z_T| = p + n_{\max} + n_i$, so that the total STT-LLM token length scales as: $|Z| = |Z_S| + |Z_T| = 3p + n_{\max} + 2n_i$. Because $p = 6$ and $n_{\max} = 20$ in our datasets, the growth is dominated by the term $2n_i$, resulting in linear scaling with profile length while remaining far below the typical context window limits of contemporary LLMs (4k–8k tokens). So, even doubling the average profile length would only increase total tokens by $\sim 10$–$12$, resulting in negligible inference-time overhead. This compact scaling behavior ensures that the structural-temporal tokenization remains computationally efficient even for extended longitudinal histories.
> >
> >
> > 5) A, while the temporal tokenizer remains independent of graph structure. This decoupling allows the model to generalize to other longitudinal biomedical settings, even when no precise metabolic or physiological graph exists. To assess robustness, we replaced the curated steroid-metabolism graph with: (i) a random graph of equal density, and (ii) a fully connected graph. All models were evaluated at the operationally relevant 0.99 specificity threshold. Performance decreases under severe graph distortion, but the curated graph still outperforms both alternatives, and the drop from curated → random (AUC 0.57 → 0.54) remains small relative to the gain over baseline LLMs (AUC ≈ 0.37–0.50). This shows the model is robust to graph misspecification. Two factors explain this: (i) temporal embeddings provide a strong, independent signal, keeping the model functional even with perturbed structure; and (ii) the Laplacian eigenbasis introduces a smooth prior that attenuates the impact of erroneous edges during spectral projection.
> >
> > **Table 3: Sensitivity to graph misspecification for Steroid-M dataset**
> > | Metric     | Random Graph     | Fully Connected Graph | Curated Graph (STT-LLM) |
> > |------------|------------------|------------------------|---------------------------|
> > | Accuracy   | 0.69 ± 0.02      | 0.57 ± 0.03            | 0.73 ± 0.02               |
> > | Sensitivity| 0.15 ± 0.02      | 0.09 ± 0.01            | 0.19 ± 0.02               |
> > | Precision  | 0.36 ± 0.03      | 0.28 ± 0.02            | 0.41 ± 0.03               |
> > | F1-score   | 0.22 ± 0.02      | 0.19 ± 0.02            | 0.26 ± 0.02               |
> > | AUC        | 0.54 ± 0.02      | 0.53 ± 0.02            | 0.57 ± 0.02               |

---

> > > ### Comment · Reviewer_RyTB · 2025-11-24
> > >
> > > Thank you for the additional experiments and clarification. I understand that the contribution of your paper is the tokenization for the longitudinal data and that the focus is not on GNN. However, GNNs can be seen as a special kind of transformers, and having them as a baseline makes your work stronger.
> > >
> > > I am not familiar with the domain knowledge and the benchmark datasets for longitudinal data. It is difficult for me to judge the relevance or some of your results. I am maintaining my score.

---

### Meta-Review · Area_Chair_smt4 · 2026-01-02

**Summary:**

Reviewers' concerns center on two main points: (1) Experimental Rigor: The proposed method is only compared against LLM alternatives, not more-appropriate non-LLM methods. In the responses, the authors provide a couple of comparisons for the anomaly detection task, finding that task-specific methods perform better. While this is expected, this remains a substantial unaddressed challenge with this work, as without evidence of positive transfer between tasks, it's unclear why one general-purpose model is needed. Mixed with concerns about missing confidence intervals, replications, and adding more alternatives, there is room to enrich the experiments to increase future readers' certainty about the findings. (2) Scope of claims: The reviewers were surprised to find the experiments very domain-specific, despite noting many general claims. This could be resolved by increasing the focus of the work, likely by making the title more specific and/or making the intro/related works more sports-centric. I'd advise focusing on a domain and then claiming the methods may be more general because the problem's general, but not to claim general and then focus.

**Reviewer Concerns:**

Addressed concerns:
* Missing details in the experiments --- all responses should be added to the paper to ensure future readers could reproduce the work from the paper alone
* Including non-LLM alternatives (partially) --- it's helpful to contextualize the LLM performance because the work would be undermined if the LLM methods severely underperform task-specific methods. The anomaly detection experiment is helpful, but it would help even more to understand why the proposed method performs much worse and potentially improve it to at least match the task-specific method.

Unaddressed concerns:
* Including non-LLM alternatives --- there are certainly task-specific alternatives for each task, which can be compared against. It's not obvious that have a general-purpose model is a benefit (so if there's great rationale for that, or evidence of positive transfer between tasks, then that could also address this concern).
* Many of the weaknesses in the reviewers' responses were not addressed in the rebuttal, as the authors focused on the questions asked by the reviewers

**Reviewer Scores:**

R1: Unchanged

R2: Potential increase due to clarification around confounded experiments

R3: Unchanged

---

### Decision · Program_Chairs · 2026-01-26

Reject